# Exogenous Estrogens as Breast Cancer Risk Factors: A Perspective

**DOI:** 10.3390/cancers17162680

**Published:** 2025-08-18

**Authors:** Parth Malik, Tapan Kumar Mukherjee

**Affiliations:** 1School of Chemical Sciences, Central University of Gujarat, Gandhinagar 382030, Gujarat, India; parthmalik1986@gmail.com; 2Swarrnim School of Sciences, Swarrnim Startup and Innovation University (SSIU), Gandhinagar 382420, Gujarat, India; 3Amity Institute of Biotechnology, Amity University, Kolkata, Plot IIA-36, 37 & 38, Action Area II, Kolkata 700156, West Bengal, India

**Keywords:** breast cancer, endogenous estrogens, exogenous estrogens, oral contraceptive pills, hormone replacement therapy, pharmaceutical estrogens, xenoestrogens, phytoestrogens

## Abstract

Estrogens are naturally occurring steroid hormones produced by the human body serving multiple physiological roles, including the development of primary and secondary sexual organs. However, the human body can also be exposed to various environmental estrogens or estrogen-like compounds, such as different pharmaceutical estrogens, xenoestrogens, and phytoestrogens. Many of these external estrogens compete with natural estrogens to bind to ERs, mimic their biological effects, and disrupt the physiological events of natural estrogens. Consequently, exogenous estrogens influence the formation and bioavailability of endogenous estrogens through feedback mechanisms and are involved in the development of breast cancer (BC). Currently, exogenously administered pharmaceutical estrogens and xenoestrogens are considered significant risk factors for BC due to their endocrine-disrupting properties and potential role in cancer development in various organs. In contrast, numerous studies have suggested potential benefits of phytoestrogen intake, although some studies have contradicted these results. A comprehensive understanding of the mechanisms underlying phytoestrogens’ actions could enable more precise dietary strategies for cancer prevention. Conversely, since exogenous pharmaceutical estrogens and xenoestrogens may increase BC risk, their use in medical recommendations and exposure through industrial or agricultural sources should be strictly controlled. New compounds with estrogen-like structures must be screened for potential estrogenic activity related to cell proliferation and gene transcription. Currently, further research is necessary to fully understand the complex relationship between environmental estrogens and BC risk, with an emphasis on developing predictive caution. With such insights, this comprehensive review article explores the role of exogenous estrogens in the development and progression of BCs.

## 1. Introduction

Estrogens are natural steroidal hormones produced in the human body that are essential for maintaining both reproductive and non-reproductive physiological functions. Under normal physiological conditions, both males and females produce estrogens. However, females have significantly higher estrogen levels due to the increased expression of aromatase, the enzyme that converts testosterone to estrogen. While ovaries are the primary source of estrogen in premenopausal women, adipose tissue also contributes significant amounts of estrogen in pre- and postmenopausal women [1]. Besides adipose tissue, the local, tissue-level expression of aromatase in various non-gonadal organs, such as the adrenal glands, bones, and brain tissues, also produces small amounts of estrogen in both pre- and postmenopausal women [2,3]. Estrogens regulate several reproductive and non-reproductive physiological functions and are vital to human health. Figure 1 depicts the major reproductive and non-reproductive functions of estrogens in the female body. Table 1 describes the comparative reproductive and non-reproductive roles of estrogens in females versus males. The diversity of controlled functions, elapsing bone health, brain functioning, skin, liver, and reproductive cycle regulation decipher estrogen’s significance for a healthy functioning of body organs.

Besides forming endogenous estrogens, the human body is exposed to various exogenous sources, such as pharmaceutical estrogens, environmental xenoestrogens, and food-derived phytoestrogens. The US Food and Drug Administration (FDA) and the global medical community have largely approved the use of estrogens for premenopausal and postmenopausal women for various purposes. Different doses of natural or synthetic estrogens are given to premenopausal women, either as oral contraceptives or combination pills, or for treating hypogonadism, significant ovarian insufficiency, and moderate acne vulgaris. Likewise, postmenopausal women may take estrogens as hormone replacement therapy (HRT) to manage hot flashes, night sweats, vulvovaginal atrophy, dyspareunia, and osteoporosis [4]. Furthermore, hormonal stimulation in transgender individuals and in vitro fertilization involves the use of exogenous estrogens [5,6,7]. Overall, 17β-estradiol (also known as estradiol/E2), equine estrogens, and 17α-ethinyl estradiol (also known as ethinyl estradiol/EE) are the main components of HRT. While equine estrogens come from pregnant mares’ urine, EE is a synthetic estrogen. EE, mestranol, and estradiol/E2 are used either alone or with progesterone in contraceptive pills. Similar to EE, mestranol is a synthetic estrogen and is considered an EE prodrug. Therefore, combining mestranol with EE increases the foreignness of their functional synergy and may lead to side effects if dosages are altered. Besides HRT, oral contraceptive pills also mainly contain EE as the primary estrogen source. For other conditions such as hypogonadism, primary ovarian insufficiency, acne vulgaris, hormonal stimulation in transgender therapy, and in vitro fertilization, estradiol/E2 is used [7]. A lot of controversy exists regarding the benefits of estrogen therapy, either alone or in combination with progesterone [8]. Besides age, numerous complex factors, such as dose, timing, intracellular signal transduction, tissue complexity, the expression of multiple transcription factors, and estrogen receptors (ERs), determine whether estrogens have beneficial or harmful effects [9].

Environmental contaminants, pesticides, plastics, and personal care products also contain estrogen-like compounds known as xenoestrogens. These non-steroidal compounds mimic estrogens and enter the human body through inhalation, ingestion, and skin contact. Some plant-based foods, including soy products, various vegetables, and flaxseeds, contain phytoestrogens and resemble natural estrogens in structure and function [10].All natural and synthetic estrogens bind to ERs, a key step in both genomic and non-genomic estrogen actions. Additionally, many diversified estrogen-like environmental compounds, including xenoestrogens and phytoestrogens, compete with natural estrogens for binding to ERs, often at low levels, as reported in many studies. These compounds can interfere with the normal functions of estrogens in various organs, including the reproductive system, breasts, lungs, kidneys, pancreas, and brain, and are considered endocrine disruptors [9,11] and pose a risk for cancer development in organs including the lungs, breast, ovaries, and uterus [11,12,13]. A substantial body of research has examined the impact of phytoestrogens on human health, especially their role in cancer development and progression. Isoflavones, coumestans, and lignans are the main classes of phytoestrogens. While most studies indicate that phytoestrogens have protective effects against chronic disorders, including breast cancer, some studies challenge this view [14,15].

Estrogens undergo extensive modifications, primarily in the liver, through hydroxylation by several isoforms of cytochrome P450 (CYP) enzymes, namely, CYP1A1, CYP1A2, CYP1B2, and CYP3A4. These enzymes catalyse the estradiol in 2-, 4-, and 16α-positions, generating 2-hydroxyestradiol (2-OH-E2), 4-hydroxyestradiol (4-OH-E2), and 16α-hydroxyestrone (16α-OH-E1).The catechol estrogens are further oxidized to form DNA-damaging semiquinones and quinones. Additionally, following CYP-mediated hydroxylation, estrogens can undergo further metabolism through glucuronidation, sulfation, and O-methylation [16]. Natural estrogens and some of their endogenous metabolites, such as 2-OH-E2 and 4-OH-E2, can induce mutations in epithelial cells and exhibit carcinogenic effects under certain conditions [17,18]. In addition to genetic alterations (e.g., BRCA1 and BRCA2 mutations), epigenetic changes and various environmental factors influence estrogen’s physiological and pathological actions. Estrogens, their metabolites, and androgens not only contribute to the development of breast cancers but also are linked to ovarian, uterine endometrial cancers in women, prostate cancer in men [19,20], and lung cancers in both sexes [12]. Furthermore, estrogen receptors (ERs) and estrogen-related receptors (ERRs, a group of orphan nuclear receptors related to estrogen actions) also play roles in cancer development [12]. Epidemiological, clinical, and experimental studies suggest that besides natural and synthetic estrogens, long-term exposure to high levels of xenoestrogens may promote the growth and progression of breast, ovarian, and endometrial cancers [11,21,22]. Regarding their genomic actions, estrogens bind to their receptors, translocate to the nucleus, recognize and bind to estrogen-responsive elements (EREs), and activate the transcription of genes involved in cancer development and propagation [23]. However, 10–15% of breast cancer cases lack ERs, PRs, and HER2 (TNBCs), meaning estrogens are not involved in the development or complications of TNBCs [24]. Clinically, while estrogen-synthesizing aromatase enzymes and ERs are targeted in estrogen-ER-dependent cancers, other therapeutic strategies such as chemotherapy, targeted therapy, and immunotherapy are used to treat TNBCs [25]. This comprehensive review explores the role of various exogenous estrogens, including pharmaceutical estrogens, xenoestrogens, and phytoestrogens, as potential risk factors for breast cancer. Figure 2 summarizes the methodology adopted for the literature search and selection of discussed studies. The selection was optimized on the basis of correlating structure–function aspects, dosage-based BC affliction with respect to estrogenic and anti-estrogenic impact, and the investigation of human populations in different geographies.

## 2. Structure–Function Relationship of Various Natural and Synthetic Estrogens

Under normal physiological conditions, the human body synthesizes different forms of estrogen, including estrone (E1), 17β-estradiol (E2), estriol (E3), and estetrol (E4), with E2 being the most abundant form during reproductive years in females. Figure 3 illustrates the chemical structures of these estrogens, where E2 is identified as the reduced form of E1, with the double-bonded oxygen replaced by an –OH group. Similarly, E3 and E4 are distinguished by increasing numbers of –OH groups and higher alkalinity in sequence. Structural analysis shows E1, E2, E3, and E4 possess one, two, three, and four hydroxyl groups, respectively. The phenolic hydroxyl group present at the C3 position of the A ring of 17β-estradiol (the major physiological estrogen in female circulation) is responsible for estrogenic activity. This hydroxyl group can be oxidized either by accepting an electron or by losing a proton. Thus, under some conditions, estrogens might act as a pro-oxidant, leading to complications of various disease conditions, namely, cancers [26].

While E3 and E4 are the main estrogens formed during pregnancy, E1 is formed after menopause. All the natural estrogens undergo extensive metabolism through phase 1 (E1, E2, E3) and phase II reactions (E4). The phase 1 enzymes are mediated by hydroxylation, oxidation, and reduction, while the phase II reaction is primarily mediated through conjugation [17]. The major metabolites of estrogen that have carcinogenic effects are 2-hydroxy-estradiol (2-OH-E2) and 4-hydroxy-estradiol (4-OH-E2).

Figure 4 distinguishes the –OH substitution patterns of 2 and 4-hydroxyestradiols, wherein double-bond separated placements in the latter infer a higher room for H^+^ release than the former, for a stronger acidic character of the latter. The missing double-bond linkage in 2-hydroxyestradiol deciphers a devoid conjugated –OH group linkage and the corresponding shift of constituent electrons, unlike 4-hydroxyestradiol. Like 17β-estradiol, 2-OH-E2 and 4-OH-E2 also bind to ERs with different affinities. Compared to 17β-estradiol, 2-OH-E2 and 4-OH-E2 bind to ERs with a lower affinity, thereby decreasing their potency in causing BC. However, 4-OH-E2 exhibits overproduction in BC. Additionally, 4-OH-E2 is capable of binding to ERs at a low dissociation rate.

Therefore, prolonged activation of the ERs induces uncontrolled cell proliferation [27]. Figure 5 illustrates the chemical structures of EE and mestranol, with –OH in EE being modified to –OCH_3_ in mestranol, indicating increased hydrophobicity and a synthetic nature. Figure 5 depicts the stronger hydrophobicity of the mestranol through a –CH_3_ group replacement of H in the –OH functionality of EE. Both are key components of contraceptive pills, with mestranol being equated as an EE prodrug. The ethinyl group at the 17-position of EE significantly increases its estrogenic potential. The presence of the ethinyl group increases the oral bioavailability of EE when compared with 17β-estradiol. Helgason S. et al. measured the level of pregnancy zone protein/pregnancy-associated alpha 2-globulin (PZP/PAα2 G) in postmenopausal women and suggested that EE is 500 times more potent than 17-β estradiol and 650 times more potent than estradiol valerate [28].

Figure 6 depicts the structural distinctions of E2 and progesterone, wherein terminal –OH groups in the former are missing in more hydrophobic progesterone.

Both these hormones are steroids, with the prominent functions of estrogen being the menstrual cycle, maturation of the Graafian follicle for pregnancy, and development of secondary sexual traits in females. Analogous to this, progesterone prepares the uterus for implantation and sustains the pregnancy, thereby complementing E2 in the conduct of HRT. Both are involved in the development of cancer, although the role of progesterone is less clear; even some results show a positive stimulatory effect of progesterone on cancer development [29].

## 3. Pharmaceutical Estrogens as a Potential Risk Factor for Breast Cancer

Pharmaceutical estrogens are primarily used by women either as part of premenopausal contraceptive pills or postmenopausal hormone replacement therapy (HRT). Additionally, for therapeutic purposes such as managing hypogonadism, transgender hormone therapy, in vitro fertilization, and other homeostatic conditions, estrogens may be prescribed by physicians. As early as 1896, George Beatson demonstrated that removing ovaries from premenopausal women with advanced breast cancer reduced tumor size. These findings indicated the involvement of ovary-produced molecules, such as steroid hormones, in breast cancer development and complications, which was later confirmed by the reduction in tumor size following the absence of steroid hormones [30]. In 1962, the discovery of estrogen receptors (ERs) by Jensen and colleagues marked a significant breakthrough in estrogen research [31]. Later, the identification of ERs, specifically ERα and ERβ, and the cDNA cloning of their respective genes (ESR1 and ESR2) in studies by the groups of Chambon and Gustafson laid the foundation for the current understanding of estrogen signaling [32,33]. In genomic actions, estrogens bind to their receptors, forming the estrogen–ER complex, which translocates to the nucleus, recognizes and binds to estrogen-responsive elements (EREs), and activates the transcription of genes involved in cancer development and progression. While ERα is recognized as being responsible for breast cancer development, the role of ERβ is controversial [23].The interaction of estrogens with ERs is highly complex, and several other signaling molecules are involved in ER-dependent and ER-independent breast cancer development and complications, including TNBCs [24].Notable signaling molecules and pathways that are involved in the complex interplay of ERs in estrogen-dependent or -independent breast cancer development and progression are growth factors, cytokines, and TGF-β1. Additionally, the extracellular matrix (ECM) components and the dynamics of interaction with the ERs emerge as a pivotal factor in modulating the epigenetic mechanisms of breast cancer complications [34].

The use of pharmaceutical estrogens as oral CPs or HRT constituents may be associated with potential side effects, including BC development and progression. Studies describe HRT as a BC risk factor, with estrogen type, typical HRT dosage and duration, subject’s age, including overall health status (pre- versus postmenopausal), and prior BC exposure as decisive aggravating factors. Several levels of evidence indicate that the genetics of the subject and varying environmental factors together determine HRT’s rationality as a BC risk factor. On the other hand, oral CPs being used by sexually active premenopausal women could aggravate the BC risk. However, controversy exists regarding the potential contribution of various HRT estrogenic components and contraceptive pills as BC risk factors.

### 3.1. Role of Hormone Replacement Therapy as a Potential Risk Factor for Breast Cancer

In HRT, estrogen is used either as a mono (estrogen-only therapy) or combination therapy (jointly with progesterone). Women who have had a hysterectomy are treated with estrogen monotherapy. Combination therapy (estrogen plus progesterone) is used in postmenopausal women with an intact uterus to alleviate postmenopausal symptoms, including osteoporosis. Estrogen and progesterone are administered simultaneously or separately as a continuous HRT or sequential/cyclical HRT through various routes, including oral intake or transdermal patches, gels, or subcutaneous implants.

Additionally, low-dose tropical HRT can be administered as vaginal creams, tablets, and rings without increasing the BC chance [35]. The Nurses’ Health Study (NHS)and its 10-year follow-up after adjusting for age and other risk factors claimed that the overall relative risk of major coronary disease in women currently taking estrogen was 0.56 (95 percent confidence interval, 0.40 to 0.80). The results indicate thatthe risk of coronary diseases was significantly reduced among women with either natural or surgical menopause [8]. Several studies have claimed that pharmaceutical estrogens used in hormone replacement therapy (HRT) and oral contraceptive pills increase breast cancer risk. Conversely, other studies argue that progesterone, not estrogen, in contraceptive pills is what elevates this risk. An eminent Women’s Health Initiative (WHI) study claimed that systematic estrogen monotherapy does not increase BC risk, and perhaps, in certain conditions, it may even lower BC risk. However, systemic combination therapy increases BC risk, even on 10 years of terminated HRT use in females. Additionally, higher doses of combined HRT (estrogen and progestins) confer a greater BC risk viarelapse in advanced BC patients [36,37]. In a WHI randomized trial, estrogen + progesterone therapy and family history were screened as independent (non-interacting) factors toward invasive BC risk among sufferers [38].

A large amount of conflicting data is available regarding the usefulness of estrogen monotherapy, besides estrogen and progesterone combined therapy, as HRT. For example, some studies revealed a small BC risk in estrogen monotherapy in the form of HRT. In contrast, others noticed no increased risk and rather revealed even minor protection by estrogen-only therapy. Also, estrogen-only therapy as HRT increased the chances of ovarian and uterine cancers in patients with intact uteri. In contrast to the general incidence of increased BC risk in estrogen + progesterone therapy, several observational studies correlate this combined therapy with a favorable BC prognosis. Controversy also exists regarding the impact of HRT on women with prior BC exposure [39,40,41]. Collectively, evidence from observational studies, randomized trials, and meta-analyses of the effects of HRT on BC’s recurrence and survival remains controversial [42].

### 3.2. Oral Contraceptive Pills as a Potential Risk Factor for Breast Cancer

Controversy exists today regarding the exact impact of oral CPs on BC development or complications. In the last century, several studies of repute examined the relationship between the use of oral CPs and BC risk. The prominent efforts in this direction are the Cancer and Steroid Hormone Case-Control (CASH) study and the Women’s Contraceptive and Reproductive Experiences (Women’s CARE) study. The findings revealed no association between the use of oral CPs (current or former use) and BC risk [43,44]. However, in another systematic review from 1996, a meta-analysis of the pooled results from 54 epidemiological studies observed a modest increase in BC risk for current oral CP users. Analysis revealed a gradual moderation of relative BC risk on terminating oral CP use [45]. In this study, the results of the meta-analysis mentioned that for women who were taking combined oral contraceptives in the 10 years after initial termination there is a small increase in the relative risk of having BC diagnosed (relative risk [95% Cl]: in current users—1·24; 1–4 years after stopping—1·16; 5–9 years after stopping—1·07. The meta-analysis further mentioned that there is no significant excess risk of having BC diagnosed 10 or more years after stopping use (relative risk 1·01).Through another significant effort, *Zhu and colleagues* meta-analyzed the data from 13 prospective cohort studies involving 11,722 cases and 859,894 participants. The analysis revealed an insignificant rise in BC risk associated with withdrawal of CP use. However, the researchers analyzed select studies involving long-term, oral CP users and claimed a higher risk of BC development [46].Several other studies also examined the relationship between the current or former use of oral CPs and breast, cervical, colorectal, and endometrial cancer risks in women of different age groups, including premenopausal women. Meta-analysis revealed a slightly increased risk, ranging from 8 to 24% [47,48]. However, a selected pre-2010 systematic review and meta-analysis of a case-control study revealed that the oral CP does not aggravate BC risk but rather its use for more than 5 years or before a first full-term pregnancy can modify BC development [49]. In an in vitro study, the treatment of MCF-7 BC cells with EE, an active component of CPs, increased their number by activating proliferation, survival, and preventing apoptosis [50]. In general, although most studies revealed no significant rise in BC risk after oral CP use, further studies are needed for claims about the therapeutic usefulness of oral CPs.

## 4. Environmental Xenoestrogens as a Potential Contributing Factor for Breast Cancer

Xenoestrogens are a group of external chemical compounds created through human-made processes and industrial activities. They enter the human body through inhalation, skin contact, and the ingestion of contaminated water, food, or supplements. Major sources of estrogens include personal care products, plastics, industrial byproducts, pesticides, herbicides, and by-products from cooking under high heat (Figure 7). The main exposure routes are dermal absorption (from personal care products and industrial environments), inhalation (airborne particles and volatile organic compounds), ingestion (food and water), and vertical transfer through the placenta and breast milk [51]. These diverse compounds are broadly categorized as synthetic estrogens (e.g., d or DES), alkylphenols (e.g., 4-nonylphenol), phenyl derivatives (e.g., bisphenol A or BPA), and organochlorines (e.g., DDT, PCBs, parabens like 2-ethylhexyl 4-hydroxybenzoate), all exhibiting different levels of affinity for estrogen receptors [52]. Various demographic factors that influence the exposure and effects of xenoestrogens on human health are genetic predisposition, age, and sex of the individual. The other demographic factors of influence on xenoestrogen exposure are geographic locations, lifestyle choices, dietary habits, occupational factors, and socioeconomic status. Several population-based studies have been conducted to show a positive association between serum total xenoestrogen levels and breast cancer risk. In one such study, Barriuso et al. showed the importance of evaluating mixtures of EDCs, rather than single chemicals, in epidemiological studies on hormone-related cancers [53]. In 2020, *Zeinomer N and groupmates* conducted a systemic review using 100 publications on epidemiological studies on environmental chemical exposures (ECEs)and breast cancer risk in women. Their results concluded that women at higher BC risk through family history, younger age of onset, and genetic susceptibility consistently support an association between an ECE and breast cancer risk [54].

In a recent meta-analysis study of 67 publications, *Liu and colleagues* showed a statistically significant association of several EDCs (organochloride pesticides, PCBs, Phthalates, diesters and their metabolites, PFAS, flame retardants, and BPA) as BC risk factors [55]. Figure 8 depicts the chemical structures of select xenoestrogens, describing their hydrophobicity-driven hampered biodegradation-dependent aggressive reactive tendencies.

Notably, the hydrophobic chains of 4-nonylphenol and 2-ethylhexyl-4-hydroxybenzoate impair their water solubility and hamper their natural biodegradation. Due to these reasons, they are likely to enter food chains and food webs, accumulating pollutant manifestation at each successive stage. The befitting case of DDT harbors familiarity, wherein 6 chlorine atoms in close vicinity, impart an enhanced reactivity and persistent tendency. Regulatory guidelines have even restricted the use of this compound as a pesticide due to its environmental impact.

The xenoestrogens are highly distinct from a structural viewpoint, all having common lipophilic-phenolic rings and other hydrophobic components, a characteristic they share with steroid hormones and their nuclear receptor-activating compounds. Such a large number of varied compounds bind to endocrine receptors, including ERs, and mimic select functions of endocrine molecules. However, xenoestrogens are weakly estrogenic, and their activity is manifold less than that of natural physiological estrogen, i.e., E2. Synthetic and natural xenoestrogens potentially modulate the effective functions of the endocrine system even at low concentrations [56]. Figure 9 depicts the major molecular targets through which estrogens interact to gain entry within the physiological boundaries. The inherent ability to bind to endocrine receptors leads to familiar xenoestrogens as endocrine-disrupting compounds (EDCs), which interfere with the formation, trafficking, and metabolic functions of endocrine molecules. The US-EPA (Environmental Protection Agency) recognized xenoestrogens as potential exogenous agents that interfere with the homeostasis, propagation, and sustenance of various endocrine agents, including estrogens [22,57], mainly about aggravation of BC and other diseases [53,58]. Several reports indicated that in an individual’s lifetime, exposure to xenoestrogen(s) is the deciding factor for their adverse or protective effects [59]. A comparative docking study of estrogens and xenoestrogens with ERs shows that both estrogens and xenoestrogens are capable of binding to the ERs, although with different affinities. Most of the xenoestrogens bind the ERs at a low affinity. When bound with the xenoestrogens, the ERs show a different conformation alteration compared to estrogen binding to ERs. Several studies have developed molecular modeling methods to study comparative docking between natural estrogens and xenoestrogens [60,61]. While working through ERs, xenoestrogens can complicate breast cancer in the absence of ERs (in TNBC cells), by interacting with other receptors such as G-protein coupled estrogen receptors (GPERs), estrogen-related receptor gamma (ERRγ), and others [59]. Additionally, xenoestrogens can affect the tumor immune microenvironment (TIME). Thus, therapeutic strategies of both hormone-dependent and independent cancers are influenced by exogenous xenoestrogen concentrations and the time of exposure [11].

Several xenoestrogens that affect human health and physiology originate from pesticides, plastic containers/food cans/beverages, and personal care items. Traditionally, it is believed that the adverse effects of xenoestrogens are proportional to their doses, and no adverse effects are seen at exposures below the LOAEL (lowest observed adverse effect level) [62]. However, different hazard assessment endpoints are studied in low-dose versus high-dose research, and a non-monotonic dose-response (NMDR) curve is created [63,64]. These findings strongly suggest that not only a single, but also a mixture of XEs at low doses can significantly affect cell proliferation, apoptosis, and tissue development even at extremely low concentrations (e.g., picomolar) [56].

DDT and its metabolites, including endosulfan, dieldrin, and atrazine, are some of the well-studied xenoestrogens. The plastic containers and beverage cans leach several agents, the prominent amongst which are Bisphenol A (BPA), alkyl phenols, and phthalates. Similarly, several personal care items such as plasticizers, perfume fixatives, and dibutyl phthalate contribute to xenoestrogens’ carcinogenic potential. In 2020, through a comprehensive review article, *Calaf and associates* described the endocrine-disrupting effects of select xenoestrogens on human health. The well-studied xenoestrogens with carcinogenic potential described in this paper are BPA, DDT, and PCBs, of which DDT and PCBs exhibit similar effects. These effects influence the proliferation and programmed cell death (apoptosis) [13]. Similarly, butyl benzyl phthalate (BBP), another endocrine disruptor with estrogenic effects, is reported to induce neoplastic transformation of human breast epithelial cells. The effects were similar to E2, although structural distinctions significantly infer a fair contrast [65] (Figure 6 and Figure 8f). The BBP used as a plasticizer gained prominence a decade ago. However, its use was banned due to multiple health concerns. The following paragraphs discuss the major xenoestrogens-biochemical aspects, concerning their health risks and safer usage extents.

### 4.1. Bisphenol A

Prevailing as a member of a large group of diverse xenoestrogens, BPA is polymerized to prepare epoxy resins and polycarbonate plastic. It shares structural similarities with other bisphenols, namely BPAF (bisphenol AF), BPF (bisphenol F), and BPS (bisphenol S) (Figure 10a–d). While BPA and BPF are common, except for the more substituted hydrophobic connecting linkages (Figure 10a,c). BPAF is characterized by enhanced chemical reactivity and intramolecular stress due to electronic repulsion amongst the 3F atoms (at each terminal) prevailing nearby (Figure 10b). However, BPS is distinguished by hydrophilic sensitivity of double-bonded S-O atoms in the central region, attributed to vigorous aggregation (Figure 10d). The distinctions in BPS and BPA chemical structures are the reasons correlated with a higher thermal and photo-stability of the former [66]. BPA is leached from plastic-lined food containers, beverage cans, and dental sealants, ultimately gaining exposure to humans [67,68,69]. Harboring a structural similarity with E2 and interfering in normal endocrine functions in a dose and time-dependent manner, BPA remains well-recognized as an EDC. The exposure of humans to BPA has been associated with epigenetic and genetic modifications that are potentially genotoxic and induce genomic instability. One of the most evident epigenetic modifications herein is aberrant DNA methylation, altering the actions of several genes, such as LAMP3. BPA also aggravates reactive oxygen species (ROS) generation by disrupting the mitochondrial electron transport chain, altering DNA structure, and binding, cumulatively exhibiting immunosuppressive effects.

High-level generation of ROS is well-known as the complicating factor of different cancers viaaltered redox control of signaling events [70,71]. Several lines of evidence indicate that epigenetic and genetic alterations of BPA affect BC complications. For example, BPA-induced LAMP3 is associated with crucial roles in BC development. Furthermore, BPA influences the expression of genes involved in proliferation and apoptosis of BC cells, including HOXB9, which is implicated in promoting cell proliferation and neo-vascularization. Other cell signaling effects of BPA that alter the breast microenvironment include regulation of growth factor signaling in breast epithelial cells and stimulation of stem cell differentiation [72]. On a similar line, *Boszkiewicz and colleagues* showed that post-menopausal women with high serum BPA and mono-ethyl phthalate levels have elevated breast density [73]. Of note, BPA has been demonstrated to influence *PTTG1* and *CDC20* genes, affecting the cell cycle alongside subsequent proliferation and migration of MCF-7 BC cells. Additionally, BPA decreased miR-381-3p, resulting in increased BC cell activity [74]. Other studies showed that in mammalian cells, low concentrations of BPA upregulate the c-Myc and other cell cycle regulatory transcription factors, leading to the promotion of cell proliferation. The study results further concluded that low-dose BPA exerted c-Myc-dependent genotoxic and mitogenic effects on ERα-negative mammary cells. In a noteworthy effort, *Kovacic and the accomplices* discussed a study related to BPA toxicity affecting metabolism, electron transfer, and oxidative stress [75,76,77].

Several other pro-proliferative and pro-survival transcription factors (e.g., STAT3) and signaling molecules (e.g., MAPK, PI3K/AKT) are affected by BPA in breast or ovarian cancer cells [78]. It appears that the estrogen-like action of BPA is mediated by ER-dependent and ER-independent functioning. Hence, it is presumed that BPA progresses ER^+^ and ER^-^ BC, including triple negative breast cancers (TNBCs). It is also noted that various BPA chemical analogs, such as BPAF, BPF, and BPS, show similar or stronger estrogenic and possibly carcinogenic effects. These observations suggest an important role of BPA in BC development and progression.

### 4.2. Dichloro-Diphenyl-Trichloroethane

Dichlorodiphenyltrichloroethane (DDT) is an organochlorine pesticide that is metabolized in the human body to its active metabolite, dichlorodiphenyl dichloroethylene (DDE). In 1991, the International Agency for Research on Cancer (IARC) classified DDT as a group 2B compound and rated it possibly related to cancer [79]. The exposure of a female body to EDC is linked to BC development and progression [80]. It was Wolf and his associates in 1993 who showed for the first time that DDT exposure in females aggravated BC development and progression [81]. Since then, several in vitro cell culture and in vivo animal model experiments have been conducted on BC cells to show the tumor-promoting potential of xenoestrogens, including DDT [82,83,84]. Elucidation of DDT’s mechanistic action relates to oxidative stress, proinflammatory reactions, and immunosuppressive actions. Select proinflammatory and pro-oxidative traits of DDT include the upregulation of COX-2 and HMOX-1 (pro-inflammatory enzymes), CXCL8 (a chemokine), TNFα (a pro-inflammatory cytokine), iNOS (inducible nitric oxide synthetase), prostaglandins, and NF-κB (nuclear factor-kappa B, a pro-inflammatory transcription factor [85,86,87]. Studies on DDT consumption in rodents revealed that DDT ingestion induces oxidative DNA damage, resulting in hepatocellular eosinophilic foci and hepatic neoplasia [88]. In rabbits, DDT and other similar pollutants exhibit immunosuppressive actions, impairing the antigen-induced serum-globulin levels [89]. Based on several studies, it was hypothesized that xenoestrogen exposure, mainly during growth and adolescence, plays a significant role in BC development by posing a significant risk factor [90].

### 4.3. Polychlorinated Biphenyls

According to IARC, polychlorinated biphenyls (PCBs) are a group of non-inflammable chemical compounds in plasticizers, capacitors, transformers, and pigments, considered as potent carcinogenic agents-known to aggravate BC in women [91]. Several cell culture and animal model studies have confirmed endocrine disruptors’ pro-carcinogenic potential, including PCBs [13]. Similar to the DDT functional mechanism, studies demonstrate aggravated oxidative stress, pro-inflammatory reactions, and activation of pro-inflammatory transcription factor NF-κB as PCB’s functional mechanism [92,93]. Additionally, in rodents and humans, toxic PCBs and other environmental pollutants have been demonstrated to promote immunosuppression [94,95].

### 4.4. Polycyclic Aromatic Hydrocarbons

Polycyclic aromatic hydrocarbons (PAHs) are a large group of organic compounds, structurally distinguished by two to seven fused benzene rings [96]. Some well-recognized PAH compounds are 7,12-dimethylbenz[a]anthracene (DMBA), benzo[a]pyrene, benz[a]anthracene, and dibenz[a,h]anthracene. Figure 11a–e depict their chemical structures, wherein DMBA hydrophobicity is moderated in benz[a]anthracene viamissing –CH_3_ substitution.

Except for dibenz[a,h]anthracene, all compounds exhibit a common hydrophobic moiety, suggesting their environmental persistence. The dibenz[a,h]anthracene is strikingly distinct, having 2 substituted benzene rings over a single benz[a]anthracene (Figure 11c,d). Thus, the low molecular weight, open structure, and geometric asymmetry favor prompt benz[a]anthracene biodegradation over the symmetric and high molecular weight dibenz[a]anthracene. A major source of PAHs is the incomplete combustion of fossil fuels [97]. The PAHs are generated due to the use of cooking materials (e.g., fats and proteins) by barbecuing/grilling, roasting, and smoking. The high temperature generated during these cooking procedures causes incomplete combustion or pyrolysis, which produces these hydrocarbons [98].

Like other xenoestrogens, PAHs are involved in pro-inflammatory and pro-oxidative outcomes and are known to suppress immunological reactions [99]. Studies via cell culture attempts and animal models elucidate PAHs-immunotoxicity in the primary lymphoid organs (bone marrow and thymus) as well as secondary lymphoid organs (spleen and lymph nodes) [100]. In the in vitro conditions comprising cultured MDA-MB-231 cells, PAHs exposure induces oxidative stress by activating non-phagocytic NADPH oxidase 2 (NOX 2) [101]. In the in vivo experiments using rat models, benzo[a]pyrene decreases the count of eosinophilic granulocytes, natural killer (NK) cells, lymphocytes, B-cells, and immunoglobulin generation from B cells [102]. A notable biochemical outcome of PAHs pertains to their recognition as aryl hydrocarbon receptor (AhR) ligands with a moderate to strong affinity [8,100]. A prominent response from the AhR-triggered PAH involves fastened Th17 differentiation that stimulates the IFN^+^ generating dendritic cells (DC) [103,104].

### 4.5. 2,3,7,8-Tetrachlorodibenzo-p-Dioxin

2,3,7,8-tetrachlorodibenzo-p-Dioxin (TCDD) originates from the combustion or emissions of electric power plants, iron and steel industries, and other metal manufacturing facilities [105]. This lipophilic molecule accumulates in fat tissues, contributing to electrolyte imbalances and impaired renal function. The high doses of dioxin are related to cardiac arrhythmias, gastrointestinal issues, and neurological symptoms, and therefore, dioxin is considered highly toxic [106]. Several investigations show that dioxins bind to the aryl hydrocarbon receptor (AhR) interacts with ERs, affecting multiple estrogen signaling pathways. TCDD, in particular, is known to disrupt the endocrine system, which is responsible for hormone generation and regulation. Thus, dioxin is considered an environmental xenoestrogen and shows carcinogenic activities, including being a BC risk factor [107,108,109]. The in vitro attempts also confirmed that TCDD promotes BC via suppression of apoptosis and anchorage-independent growth of the human breast epithelial cells with stem cell characteristics [110,111]. The findings were reciprocated in the rodent uterus, wherein TCDD exposure enhanced the uterus’s susceptibility to BC development [111]. Though two successive French cohort studies by the same group have revealed TCDD tumor-promoting actions through hemopoietic neoplasms, lymphatic disorders, brain and biliary complications, no such association has yet been demonstrated for BC risk [112,113].

### 4.6. Mechanism of Action of Xenoestrogens in the Development of Cancers

Exogenous estrogens, especially xenoestrogens, have enduring pathophysiological effects by either directly affecting genetic and epigenetic changes or by competing with endogenous estrogens for binding to ERs, thereby disrupting the natural hormonal signaling mediated by the estrogen-ER complex. Multiple lines of evidence indicate that both the normal physiological and pathological roles of endogenous estrogens are influenced by exogenous estrogens. This imbalance in estrogen levels caused by external xenoestrogen compounds impacts several pathological processes, including infertility, immune disorders, various cancers, and other health problems.

A thorough screening of cellular and molecular events related to the development and progression of different cancers reveals that interaction between exogenous estrogens and ERs can initiate and promote multiple cancers, including BC, through four main signaling pathways. 1. In the classical genomic pathway, xenoestrogens diffuse into the cytoplasm and bind to dimeric ERs, then translocate to the nucleus. Here, the xenoestrogen-ER complex binds to estrogen response elements (ERE) in gene promoters controlled by estrogen, regulating transcription. However, not all estrogen-responsive gene promoters contain EREs; some have regions where specific transcription factors bind. 2. In the non-classical genomic pathway, when EREs are absent, the cytoplasmic xenoestrogen-ER complex activates various transcription factors (e.g., SP1, AP1, GATA, STAT) which then move to the nucleus, bind to specific regions in gene promoters, and regulate transcription. 3. The third pathway involves membrane-bound ERs (mERs) and G-protein coupled estrogen receptors (GPERs), responsible for rapid, non-genomic actions, such as activating ion channels on the cell membrane. These receptors activate transcription factors that eventually bind to gene promoters of estrogen-activating genes to trigger transcription. 4. The fourth pathway describes how EREs in estrogen-responsive genes can be activated without estrogens, by the binding of specific growth factors (GFs) to membrane growth factor receptors (GFRs). The GF–GFR complex activates kinases that lead to phosphorylation of ERs, thus activating them and allowing their binding to EREs. Several other regulators and co-activators of EREs have been identified, including AhRs, peroxisome proliferator-activated receptors, and ROS. Additionally, when activated, ERRs, a group of orphan nuclear receptors, bind to the ERREs of specific genes and regulate their transcription. Xenoestrogen signaling activates several proto-oncogenes, turning them into oncogenes, and inactivates tumor suppressor genes along with genes involved in inflammation and oxidative stress. Key cell signaling molecules and pathways affected by xenoestrogens include those involved in cell proliferation (e.g., cyclin D), survival (e.g., AKT), apoptosis (e.g., Bcl-2), angiogenesis (e.g., VEGF/VEGFR), and invasion and metastasis (e.g., MMPs) in cancer cells. In a comprehensive review, Wang X. et al. described the impact of xenoestrogens and phytoestrogens on various cell signaling molecules and cellular events related to normal and abnormal functions [10]. A schematic diagram depicts the various mechanisms through which xenoestrogens mediate pro-carcinogenic action (Figure 12).

## 5. Phytoestrogens as a Potential Preventive Factor for Breast Cancer

Phytoestrogens are a group of plant compounds that resemble E2 and trigger binding to ERs [114,115]. The major sources of phytoestrogens are fruits (e.g., grapes, pears, plums, berries, and apples), vegetables (soybeans, sprouts, beans, cabbage, and spinach), hops, grains, onions, garlic, and beverages like tea and wine (Table 2) [116,117,118,119,120,121,122,123,124,125,126,127,128]. Additionally, People of the Indian subcontinent and Southeast Asian countries use various spices, including curry powder, for cooking, which contains several phytoestrogens.

A common structural aspect of all these phytoestrogen compounds is the prevalence of one or more phenolic OH groups, which confer their antioxidant properties. Another constant characteristic of these compounds’ functional chemistry pertains to their chemical environment and dosage-dependent anti-or pro-oxidant activity.

Studies correlate their optimum actions, including alkaline degradation, which acts as a prominent aspect of their metabolism. Isoflavones, coumestans, prenyl flavonoids, and lignans are the primary dietary sources of phytoestrogens, with potential health benefits [129,130]. One of the important phytoestrogens with potential anticancer activities is curcumin, although curcumin shows no structural relationship with estrogens or strong affinity toward ERs. Still, a few studies also indicated the ability of curcumin and select other phytoestrogens to influence the ERα translocation from the cytoplasm to the nucleus and its transactivation [131]. Several limitations are identified for the successful use of curcumin as an anticancer agent, including its capacity to behave both as an antioxidant and pro-oxidant, under certain conditions [132,133]. Additionally, isoflavones such as genistein, daidzein, and glycitein are the largest group of phytoestrogens, exhibiting variations from a missing –OH group in daidzein to an –OCH_3_ substitution in glycitein (Figure 13a–c). A comparative study on dietary patterns has revealed that people of eastern and southeastern Asian countries, such as the Japanese and Koreans, exhibit much lower BC incidence than Western/European countries [134,135,136]. Several levels of evidence indicate that phytoestrogens such as soy isoflavone genistein are associated with a lower BC risk [137,138,139,140]. As a cautionary stance, studies in a population of Canada have demonstrated that phytoestrogen consumption through food materials (e.g., soy food) during childhood and adolescence does not exert adverse hormonal effects in children or affect pubertal development and reduce the BC vulnerability in later stages (adult/old age) [141,142]. In China, results of a large population-based study showed that high soy protein or isoflavone consumption decreased BC recurrence and mortality [143].

Likewise, a Korean hereditary study on BCs observed that soy product consumption is related to a lower BC risk in BRCA mutation-inherited members in a family. This association was stronger for carriers of the BRCA-2 mutation than the BRCA-1 [144]. The mechanisms through which phytoestrogens may stimulate or inhibit BC growth remain controversial, as deciphered by the findings of multiple in vitro and in vivo studies. Therefore, a causal relationship is not yet known for the consumption of dietary phytoestrogens and BC risk in Asian and Western countries. It is assumed that several other factors, including tissue-specific expression of different ERs and the phytoestrogen (such as naringenin) concentrations, are important in BC development [145,146]. The following paragraphs describe some of these important determining factors for phytoestrogens’ role in BC complications.

### 5.1. Role of Estrogen Receptors in Phytoestrogen-Dependent Breast Cancer Complications

The structural similarities between phytoestrogens and E2 allow the former to bind ERs [115]. As per the findings of the crystallographic studies, isoflavones bind to ERs through the 4-OH group on their B ring [147] (Figure 13a,b). However, the binding affinity of genistein to ER is several-fold lower than that of E2, suggesting it is only weakly estrogenic [148]. Phytoestrogens regulate ER functions by acting as ER agonists or antagonists. While E2 harbors nearly equal affinity to ERα and ERβ, genistein, a popular phytoestrogen derived from soy products, exhibits a stronger affinity toward ERβ [139,149,150]. In a typical cell model system, the biotransformation of genistein and zearalenone (a primary toxin binding ERs and regulating infertility), substantially influences their estrogenic potency (Figure 13g) [151]. Another study by us using cultured vascular endothelial cells demonstrated that 17-epiestriol, an endogenous estrogen metabolite, and genistein, both attenuated TNFα-dependent vascular cell adhesion molecule 1 (VCAM1) expression through ERβ [152]. A similar study by Jiang and associates showed ER expression related to the varied phytoestrogen potency in humans. Analysis revealed low phytoestrogen doses supporting the ERβ binding to regulate the expression of certain genes. However, at higher concentrations, phytoestrogens bind ERα and ERβ with similar potency, which determines their mechanism of action [150]. Intriguingly, though, Zhang and accomplices observed a higher isoflavone dietary intake being associated with reduced mortality of ER^-^ BC patients, suggesting no necessity of ERs in anti-BC effects [153].

The phytoestrogens daidzein and its metabolite equol promote tumor growth to a lesser extent than endogenous estrogen [154]. The mechanism of growth-stimulating actions of phytoestrogens at low doses is mediated by their ER-binding ability [155]. In a notable effort, Shao and colleagues used both in vitro and in vivo xenograft mouse models and showed that genistein reduces cellular angiogenesis and apoptosis in both ER^+^ and ER^-^ BC cells, namely MCF-7 and MDA-MB-231 BC cells, respectively [156]. Estrogens regulate several key molecules required for S-phase entry in the cell cycle, including cyclin D1, Myc, Cdk2, Cdk4, Cdk inhibitors, and Cdc25A [157]. In the presence of estrogen, the binding of phytoestrogens (genistein, daidzein, and coumestrol) to ERs inhibits apoptosis of BC cells, without affecting cell cycle [158]. Other studies showed that flavonoids such as genistein and quercetin are fully estrogenic agonists of both ER isoforms. They have further shown that at low concentrations or concentrations sufficient to activate the transcription of the genes related to cell proliferation in an ERα-dependent manner. However, high concentrations of genistein and quercetin that can be reached through dietary consumption of a soya diet kill the cancer cells in an ER-independent manner [159]. In an athymic xenograft mouse model, by injecting the MCF-7 BC cell line, investigators observed that genistein substitutes endogenous estrogens to promote mammary tumor growth [160,161]. Besides concentration, some studies reported that biochanin A, another flavonoid, promotes ERα-positive BC cell proliferation through miR-375 activation [162], while genistein induces apoptosis and autophagy in MCF-7 BC cells by modulating the expression of pro-apoptotic factors and oxidative stress enzymes [163].

### 5.2. Role of Menopausal Status in the Effects of Phytoestrogen-Dependent Breast Cancer Complications

A few studies demonstrated the protective effects of dietary phytoestrogens against BCs, being confined to pre-menopausal and overweight women [164,165]. On the other hand, separate attempts by the *Guha and Kang* groups revealed that only post-menopausal women exhibit a reduced BC recurrence in women receiving tamoxifen therapy associated with a higher isoflavone intake. The isoflavone consumption did not interfere with the efficacy of tamoxifen [166,167]. Multiple other studies deciphered a positive correlation between phytoestrogen consumption and decreased risk of BC-related death and recurrence [145,168,169,170]. Additionally, several other studies retrieved better outcomes in women with hormone-sensitive cancer and/or patients receiving hormonal therapy with phytoestrogen treatment [166,167,168,169]. Another meta-analysis by Nechuta and colleagues collected data from three cohorts (LACE, WHEL, and SBCSS) from the United States and China, totaling nearly 9500 BC survivors. The investigators concluded that soy food consumption at ≥10 mg of soy isoflavones per day was associated with significantly reduced recurrence and mortality of BCs [170].

### 5.3. Concentration-Dependent Response of Phytoestrogens on Breast Cancer Complications

The phytoestrogen dosage remains a decisive factor in determining the role of BC complications [171,172]. The anti-BC actions of various phytoestrogens exhibit a concentration-dependent biphasic variation on ER^+^ as well as ER^-^ BC cell lines, with impaired growth at low extents being modulated to apoptosis at high extents [173,174]. Phytoestrogen-specific actions are observed since some phytoestrogens stimulate cell proliferation in an ER-dependent manner [160,173,174]. Such actions of phytoestrogens are thoroughly supported by the common structural feature of 2-OH groups substituted on a benzene ring, presenting a scope for further screening via correlating structure-function activities (Figure 13a,d–f). A dose-response meta-analysis was conducted on more than 300,000 women to study the impact of soy food intake on BC risk. Although the results were controversial, a 3% reduced BC risk was noticed per 10 mg isoflavone consumption each day [175]. Result analysis by Zhang and associates revealed enhanced survival with increasing soy isoflavone intake, although in a nonlinear proportion [169]. In a similar study, Shu and colleagues found a linear dose-response toward 40 mg per day soy isoflavone intake, but the impact leveled off thereafter [143]. Therefore, the extent of isoflavones appears to have a major role in the soy protein anti-BC actions. Further, large controlled clinical studies are required to evaluate the phytoestrogen’s anti-BC efficacy.

Table 3 summarizes the effect of prominent phytoestrogens on BC cells’ proliferation, the majority of which exhibit inhibiting impact along with their micro- to nanomolar IC_50_ extents [176,177,178]. A minor concern herein pertains to the majority of listed studies being associated with ER-positive MCF-7 cells, and this impact indeed correlates with the stronger ERα receptivity of MCF-7 BC cells. The distinct IC_50_ concentrations for different compounds decipher their characteristic tumor suppression abilities, such as 50 and 10 nM with genistein and daidzein, respectively. The micromolar IC_50_ for quercetin, naringenin, apigenin, luteolin, biochanin A, and kaempferol infers their lower anti-BC potency over genistein, daidzein, tamoxifen, chrysin, mestranol, and coumesterol, having nanomolar IC_50_.

Another important conclusion is the three-fold lower IC_50_ for TNBC cells, suggesting a lower scutellarein, genkwanin, and sinensetin toxicity than in the MCF-7 BC cells in the absence of ERs, indicating ERs are not necessary for the actions of certain estrogens. Finally, a noteworthy aspect is the BC-promoting activity of E2, the major estrogen in the female blood, on ER^+^ MCF-7 BC cells till 1 nM. Altogether, the IC_50_ variations in phytoestrogens signify the structure–activity relationships for their sustainable functioning by explicit functional group placements.

### 5.4. Possible Protection Mechanisms of Phytoestrogen-Dependent Cancer Complications

The effects of phytoestrogens, whether as a protector or a promoter of cancers, depend on several factors, including the type of phytoestrogen, the specific estrogen receptor it interacts with, the concentration of the phytoestrogens, and other variables. The potential anticancer activities of different phytoestrogens are generally categorized into the following protective mechanisms: 1. Modulating estrogen receptor activities; 2. Reducing pro-proliferative and pro-survival cell signaling pathways; 3. Decreasing estrogen production and altering its metabolism; 4. Epigenetic modifications; 5. Antioxidative and anti-inflammatory mechanisms. Several studies reported that phytoestrogens, such as genistein, have a preferential binding affinity for ERβ, which contributes to their antiproliferative effects. In ER-negative breast cancer cell lines, phytoestrogens help prevent cancer complications through epigenetic modifications, including alteration of DNA methylation and histone acetylation. The anti-inflammatory properties of several phytoestrogens, such as genistein and resveratrol, are well-established. A series of comprehensive review articles highlighted the various mechanisms through which different phytoestrogens protect against cancers, including breast cancer [179].

## 6. Conclusions and Future Perspectives

BC is the most common cancer worldwide and the second leading cause of cancer-related deaths. In premenopausal women, although the ovaries are the major sources of estrogens, adipose tissue and adrenal glands contribute small amounts of estrogens. Menopause significantly decreases estrogen production from the ovaries. However, adipose tissue, adrenal glands, and tissues such as the brain and lungs continue to produce estrogens due to the expression of the aromatase enzyme at the local tissue level. Besides natural estrogens, the human body, including both pre- and postmenopausal women, regardless of age and menstrual status, is exposed to various pharmaceutical estrogens, environmental xenoestrogens, and food-derived phytoestrogens. Many of these compounds, whether pharmaceutical estrogens, xenoestrogens, or phytoestrogens, bind to ERs with different affinities and cause diverse actions. At the physiological level, the interaction between estrogens and ERs regulates reproductive functions as well as other non-reproductive processes. Under certain conditions, natural, synthetic, or environmental estrogens can trigger neoplastic transformations in human epithelial cells, leading to cancers of the breast, ovaries, uterine endometrium, and lungs.

Therefore, when using pharmaceutical estrogens for therapeutic purposes in women, caution is necessary, and consulting a physician is strongly recommended, especially regarding dose, duration, and type of estrogen administered. Physicians’ recommendations and advice are highly essential regarding the dose of hormonal therapy (low dose versus high dose) and whether monotherapy (estrogen-only therapy) or combination therapy (estrogen + progesterone) may be used. Premenopausal women are particularly vulnerable to the effects of pharmaceutical estrogens due to the high physiological levels of estrogen produced by their ovaries. In postmenopausal women, endogenous estrogen synthesis sharply decreases, leading to symptoms such as osteoporosis, vaginal dryness, and increased risks of CVDs and other conditions. Several studies, including the Nurses’ Health Study (NHS), Women’s Health Initiative (WHI), and others, show the benefits of hormone replacement therapy (HRT) in alleviating postmenopausal symptoms. However, other research contradicts these benefits and warns against continued estrogen use because of potential cancer risks and adverse effects on the cardiovascular system. Besides natural and synthetic estrogens, the human body is exposed to environmental xenoestrogens from sources such as pesticides, plastic containers, and cosmetics, which can disrupt the physiology of endogenous estrogens and thus are regarded as endocrine-disrupting chemicals (EDCs). Precautions should be taken to minimize exposure to xenoestrogens due to their ability to cause genetic and epigenetic modifications (particularly DNA methylation and acetylation) and promote the development and progression of various cancers. Many studies suggest that procarcinogenic xenoestrogens tend to bind strongly to ERα, promoting cancer development.

In contrast, phytoestrogens, a group of diverse compounds found in foods such as soy products, may offer health benefits or even protection against the development of cancers. Diverse mechanisms of the protective action of phytoestrogens are observed, including antioxidative and anti-inflammatory actions. Additionally, a comparative study by Gustafson and his team (the discoverer of ERβ) showed that 17β-estradiol, the most abundant natural estrogen in women, has nearly equal affinity for ERα and ERβ. On the other hand, phytoestrogens like genistein, derived from soy, exhibit ~400 times greater affinity for ERβ, which contributes to their protective effects against cancers and CVDs. Other studies noted that the protective effects of phytoestrogens on human health, including their role in preventing BCs, depend on factors such as ethnicity (e.g., South/Southeast Asian people), although the majority of the study results claim that it is the specific food habits rather than genetics which takes a decisive role in the protective actions of phytoestrogens. Although several clinical studies have explored the potential benefits of phytoestrogens, conclusive evidence confirming their role in cancer prevention is still lacking. Patient-derived xenograft models and nanoparticle-dependent drug delivery are important areas of present-day research in breast cancer therapy [179,180,181,182]. More research is needed on the impact of environmental estrogens/phytoestrogens, including the effects of combination exposure, keeping in mind sex, age, and other critical factors, which would further enhance our knowledge in this area.

## Figures and Tables

**Figure 1 cancers-17-02680-f001:**
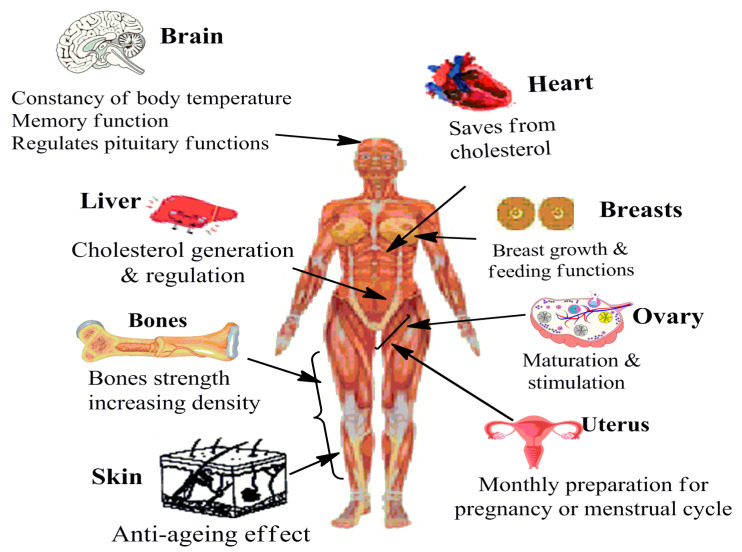
Prominent functions of estrogens in a female body system.

**Figure 2 cancers-17-02680-f002:**
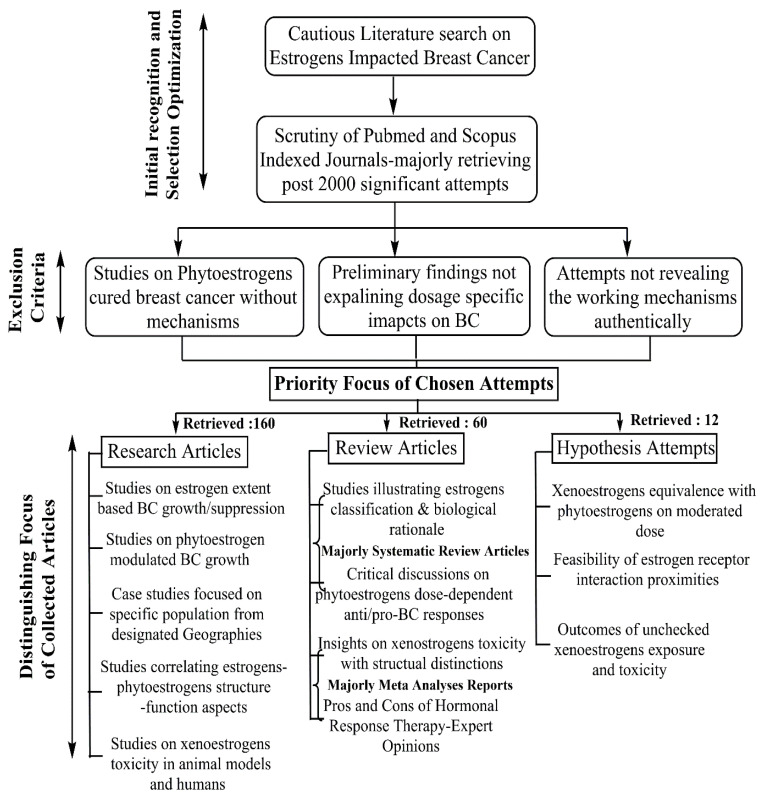
Methodology for the selection of described studies, signifying the priority selection from the available literature, among fundamental articles, case studies, and systematic reviews.

**Figure 3 cancers-17-02680-f003:**
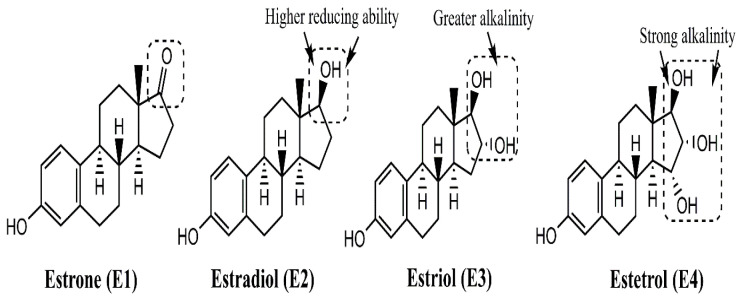
Structural distinctions of the physiological forms of estrogens, depicting the increasing –OH groups from estrone (E1) to estetrol (E4). E2 (estradiol) is the major form produced in a female body during the reproductive years, carrying fewer –OH groups than E3 and E4.

**Figure 4 cancers-17-02680-f004:**
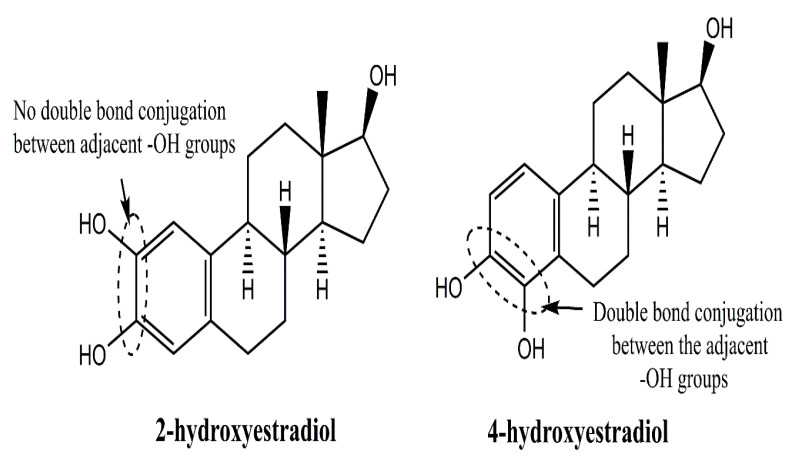
Structural distinctions of 2 and 4-hydroxyestradiol, the endogenous metabolites of estrogen, which instigate a carcinogenic response if generated in excessive amounts. The structures decipher a decisive role of double-bond conjugated-OH groups in 4-hydroxyestradiol for higher reducing ability (H^+^ release), unlike 2-hydroxyestradiol.4-OH-E2 shows a low dissociation rate from its ER complex, indicating prolonged estrogenic action.

**Figure 5 cancers-17-02680-f005:**
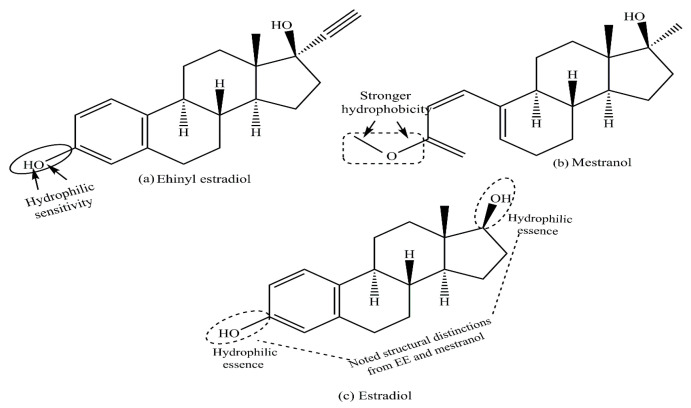
Structural distinctions of (**a**) ethinyl estradiol (EE, a synthetic estrogen used as a component of HRT and oral contraceptives), (**b**) mestranol (a synthetic prodrug of EE), and (**c**) estradiol (a major natural estrogen present in the female circulation).

**Figure 6 cancers-17-02680-f006:**
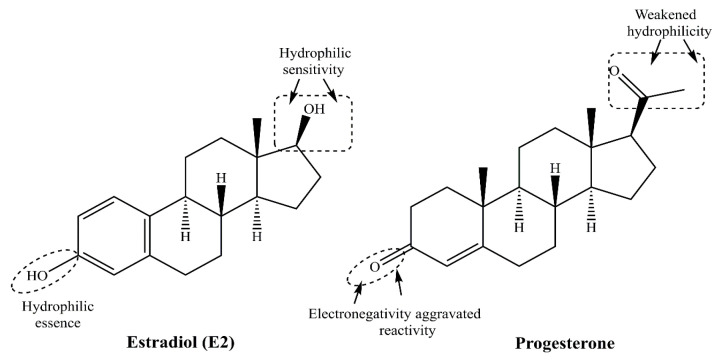
Structural distinctions of estradiol (E2) and progesterone, administered jointly as hormone response therapy ingredients to relieve postmenopausal symptoms. The figure distinguishes the moderated hydrophilicity of progesterone, with fewer –OH groups than E2.

**Figure 7 cancers-17-02680-f007:**
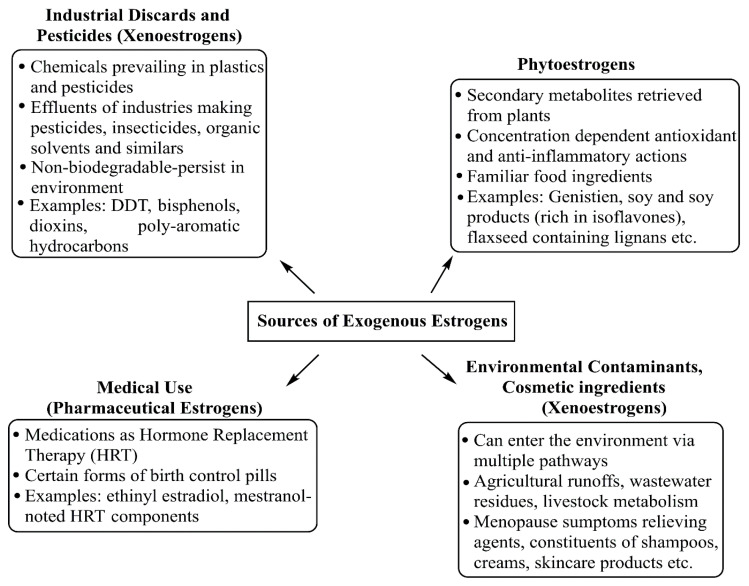
Diversified exogenous sources for estrogen afflicted environmental damage and health concerns, spanning medical use, herbal genesis, and industrial and cosmetic discards.

**Figure 8 cancers-17-02680-f008:**
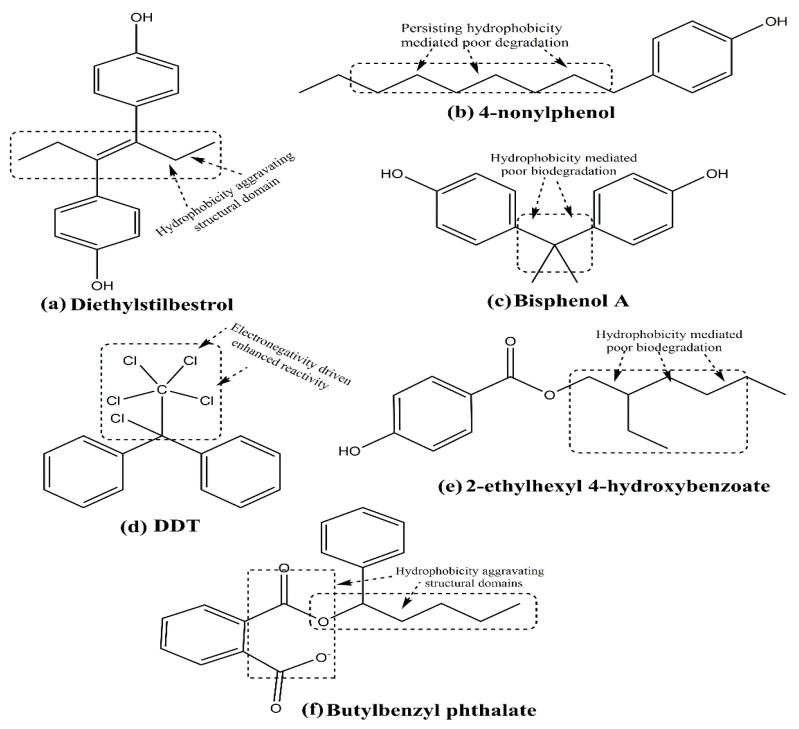
Hydrophobicity complicated biological persistence of (**a**) diethylstilbestrol, (**b**) 4-nonylphenol, (**c**) bisphenol A, (**d**) DDT, (**e**) 2-ethylhexyl 4-hydroxybenzoate, (**f**) butylbenzyl phthalate. Organic essence-mediated low water solubility is the reason for the poor natural biodegradation of these compounds, enhancing their persistence and toxicity concerns.

**Figure 9 cancers-17-02680-f009:**
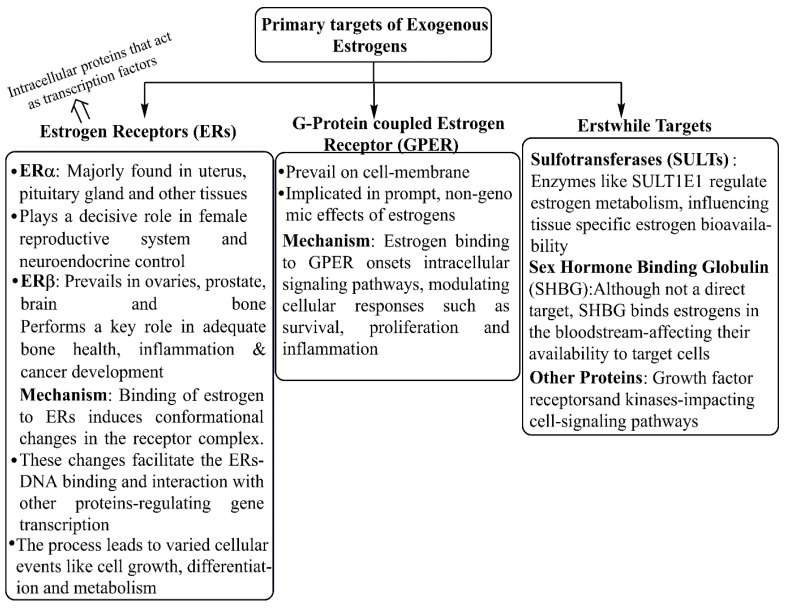
Possible interacting targets for physiological access of exogenous estrogens.

**Figure 10 cancers-17-02680-f010:**
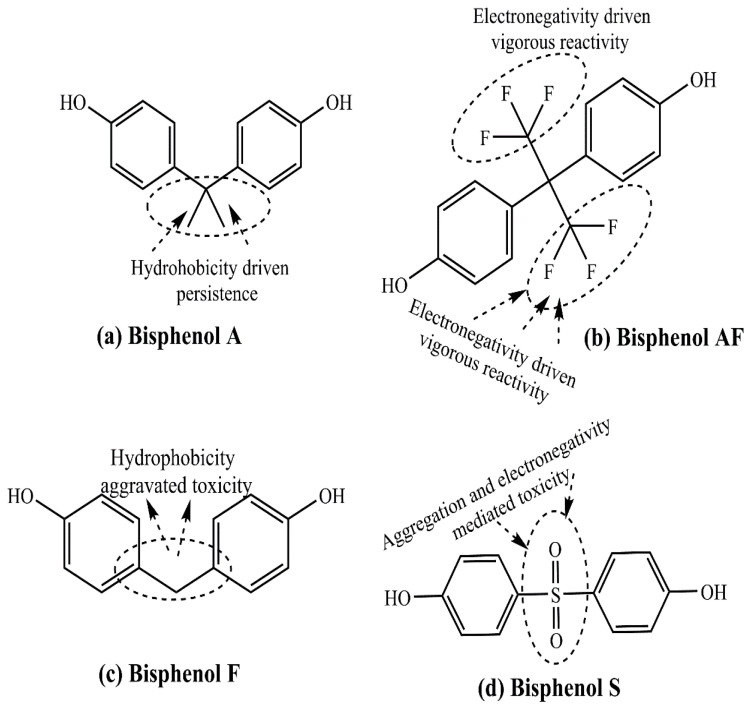
Structural distinctions of (**a**) bisphenol A, (**b**) bisphenol AF, (**c**) bisphenol F, and (**d**) bisphenol S, depicting the hydrophobicity aggravated toxicity and persisting tendencies on impaired biodegradation, causing a threat to the environment and human life.

**Figure 11 cancers-17-02680-f011:**
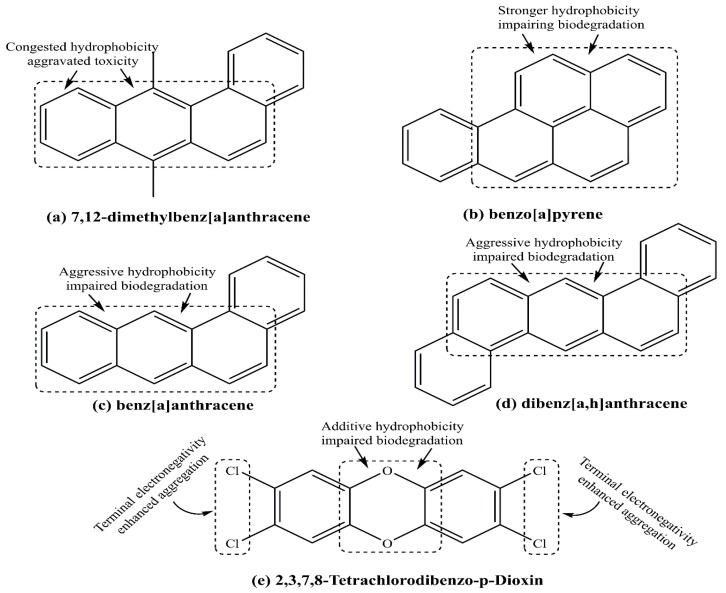
Structural distinctions of prominent polyaromatic hydrocarbons: (**a**) 7,12-dimethylbenz[a]anthracene, (**b**) benzo[a]pyrene, (**c**) benz[a]anthracene, and (**d**) dibenz[a,h]anthracene. Multiple benzene rings encompass their toxic essence, causing harm to human life via enhanced oxidative cum inflammatory stress and impaired immune response. (**e**) The terminal -Cl in 2,3,7,8-Tetrachlorodibenzo-p-dioxin alongwith the central moiety linking the penultimate phenyl rings signify its environmental persistence.

**Figure 12 cancers-17-02680-f012:**
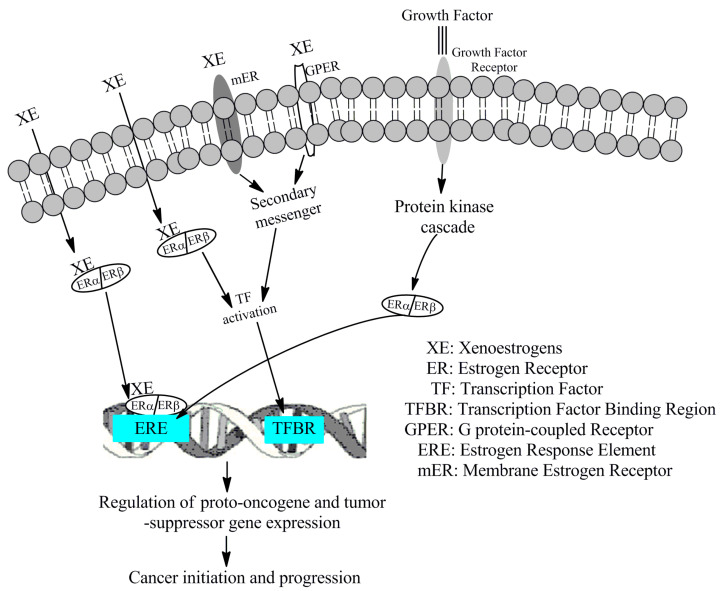
Possible signaling pathways acted upon by xenoestrogens to aggravate breast cancer pathogenesis.

**Figure 13 cancers-17-02680-f013:**
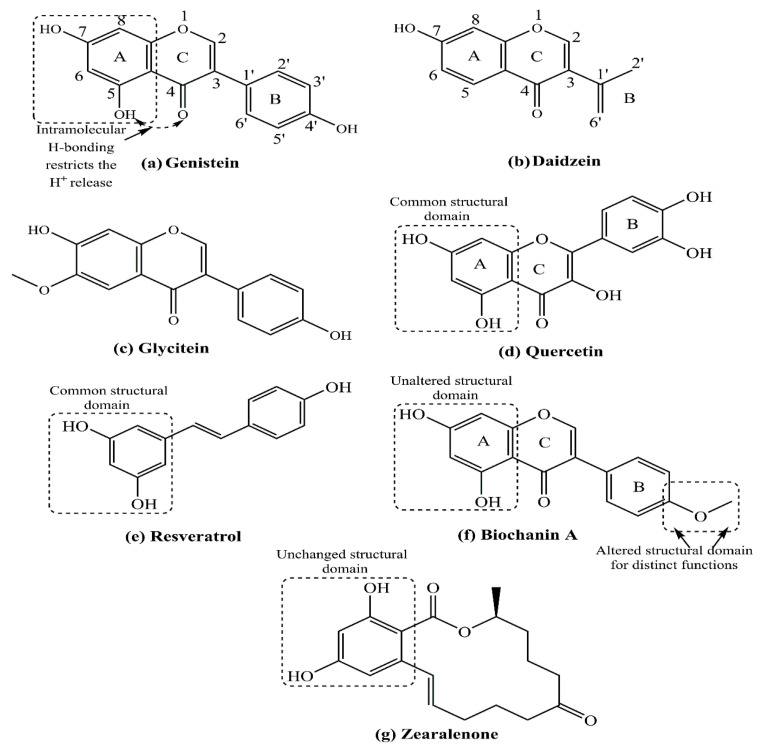
Chemical structures of prominent phytoestrogens and their derivative fractions, including (**a**) genistein, (**b**) daidzein, (**c**) glycitein, (**d**) quercetin, (**e**) resveratrol, (**f**) biochanin A, and (**g**) zearalenone. Presence of multiple –OH and O in the structure’s attributes for polyphenolic sensitivity with an H^+^ release provision mediated by electrochemical reduction abilities.

**Table 1 cancers-17-02680-t001:** Important biological and non-biological functions of estrogens in females and males, pinpointing estrogens significance for life sustenance.

Functional Specification	In Females	In Males
Reproductive	Regulation of events affecting sexual and reproductive health Rising extents during puberty assist the formation of secondary sexual organs, such as breasts and whole body variations Plays a key role in the menstrual cycle: regulating simultaneous FSH, LH, and progesterone prevalencefor regular periods Regulates ovulation alongside thickening the uterus lining for preparation of pregnancy Assists fertilization by thinning the cervical mucusfacilitating pregnancy by keeping vaginal walls thick, elastic, and lubricated	Impacts sexual functions such as erection- and sperm-generating ability Deficiency impairs reproductive functions similar to its excess-driven infertility and erectile inactions Excessive estrogen is risky too, causing enlarged breasts Assists in formation of testes and sexual desire, regulating progesterone secretion from the pituitary gland Regulates fluid reabsorption from the efferent ducts, maintaining optimal prostate conditions
Non-reproductive	Sustains important events in skeletal, cardiovascular, and central nervous systems Regulates blood sugar level alongwith bone and muscle mass Maintains blood circulation and collagen formation for skin health Low estrogen levels result in weakened bones, making them vulnerable to fracture Improper estrogen levels are attributed to endocrine and gastrointestinal disorders Involved in neuroprotection (brain functioning), DNA repair, and emotional and stress responses Regulates skin elasticity/fat distribution (conferring typical body morphology)	Estradiol deficiency impairs cognitive events, such as spatial ability and verbal memory Impacts the communication of neurons, regulating interconnections, crucial for learning and memory Estrogen deficiency can result in bone loss and increased fracture risk Influences mood and skin health Vasodilatory and guarding impacts on cardiovascular system Regulation of body weight (cholesterol metabolism) and growth of bodily hair Healthy functioning of blood vessels, fluid re-absorption, and cognitive coordination

**Table 2 cancers-17-02680-t002:** Sources and chemical distinctions of select phytoestrogens [116,117,118,119,120,121,122,123,124,125,126,127,128].

Sr. No.	Potential Compounds	Distinctive Class/ Family	Major Sources
1	Catechin, epicatechin, epigallocatechin 3-gallate	Flavanols	Green and black tea, cocoa
2	Genistein and daidzein	Isoflavones	Soy-based foods
3	Quercetin, kaemferol	Flavonols	Quercetin: Red wine, grapefruit, onions, apples and black tea Kaemferol: Broccoli, apples, strawberries, beans, tea and tomato
4	Resveratrol Pterostilbene	Stilbenoids	Resveratrol: Food and beverages derived from grapes, mulberries and peanuts Pterostilbenes: Blueberries
5	Hesperetin and naringenin	Flavanones	Citrus fruits
6	Curcumin	Curcuminoids	Rhizomes of the plant *Curcuma longa* Linn
7	Caffeic acid, ferulic acid	Hydroxycinnamic acids	Fruits, vegetables, tea, cocoa and wine
8	Pinoresinol	Lignans	Fiber-rich food, such as sesame seeds
9	Luteolin and apigenin	Flavones	Cereals and herbs
10	Esculetin, psoralen, esculin	Coumarins	Sweet woodruff, meadowsweet, sweet grass and Melilotus
11	Phenylacetic acids	Phenolic acids	Acidic tasting fruits
12	Phloridzin	Chalcones	Citrus fruits and apples
13	Vanillin	Benzoic aldehydes	Vanilla

References in caption are in order of studies mention.

**Table 3 cancers-17-02680-t003:** In vitro effects of select estrogens on breast cancer cell proliferation [176,177,178].

Sr. No.	Screened Compound	IC_50_ Extent, Cell Line Studied Upon, Water Solubility	Prominent Findings	Limitations
1	Genistein, a phytoestrogen	50 nM, MCF-7 <1 μg·mL^−1^ water solubility	Weaker hERβ binding activity than DES, EE, TAM, clomiphene, and 4-nonylphenol, bisphenol A, 4-dihydroxybiphenyl Suppression of environmental estrogen growth stimulating actions Lesser maximum estrogenic activity (90%-1 μM) over daidzein (100%-10 nM) and coumestrol (100%-100 nM) Maximum inhibition of DES (338 ± 8) and 4-nonylphenol (56 ± 10) Up to 50% impairment of 10 nM industrial chemicals induced MCF-7 cell proliferation	No conclusive screening in control (normal) cells Analysis in only MCF-7 cells leaves the results in jeopardy for TNBC cells (ER negativity) Used concentration is higher than estrogenic extents of ~2.5 nM (in various phases of menstrual cycle) No in vivo screening leaves the findings infeasible for clinical screening
2	Daidzein, a phytoestrogen	10 nM, MCF-7 0.00831 mg·mL^−1^ water solubility at 25 °C	Maximum inhibition of DES (cell number: 340 ±10) and 4-nonylphenol (56 ± 10) Weaker activities over ES (estrone) and DES Moderate suppression of EE estrogenic activity but impaired mestranol and clomiphene actions Stimulated the proliferation on par with ES but suppressed environmental estrogen actions 100 nM exhibited a pharmaceutical estrogen equivalent impairment	No conclusive screening in control (normal) cells Analysis in only MCF-7 cells leaves the results in jeopardy for TNBC cells (ER negativity) Used concentration is higher than estrogenic extents of ~2.5 nM (in various phases of menstrual cycle) No in vivo screening leaves the findings infeasible for clinical screening
3	Biochanin A, a natural phytoestrogen	10 μM, MCF-7 Not very soluble in water	Weakest estrogen binding over pharmaceutical estrogens 30% as maximum estrogenic activity at 100 μM Selective proximity to ERβ: regulating bone health, brain functions, and cardiovascular actions Modulates cortisol and androstenedione actions in certain cells Potent anti-inflammatory essence	Used concentration is higher than estrogenic extents of ~2.5 nM (in various phases of menstrual cycle) No in vivo screening leaves the findings infeasible for clinical screening May adversely affect fertility and reproductive functions Very weak estrogenic activity compared to E2
4	Quercetin, a phytoestrogen	5 μM, MCF-7 Water solubility of 0.00215 g·L^−1^ at 25 °C	Weaker estrogenic activities even at 50 μM over phloretin’s (a chalcone) at 7 μM Highest estrogenic activity (120%) at 1 nM but strong proliferation suppression at >10 nM No significant decrements in pharmaceutical and environmental estrogen activity On being applied with environmental estrogens, 10 nM extent exhibited little suppressed estrogenic activity	Used concentration is higher than estrogenic extents of ~2.5 nM (in various phases of menstrual cycle) No in vivo screening leaves the findings infeasible for clinical screening Weaker actions make quercetin a less potent anticancer agent over its other family members
5	17β-estradiol (E2), a natural estrogen	10^−2^ to 1 nM, MCF-7 3.1–12.96 mg·L^−1^ water solubility	A steroid hormone, most potent form of estrogen in humans Crucial role in regulating physiological processes, including female reproductive cycle Major impact on energyhomeostasis and nervous system Binds to ERα and ERβ with same affinity, the relative expressions leading to distinct tissue impacts Deficiency or excess availability both are detrimental	HRT involving E2 can aggravate the risk of strokes and cardiovascular diseases Estrogenic activity could be affected by environmental contaminants Prolonged exposure of E2 may aggravate oxidative stress in certain tissues Unchecked exposure may aggravate hormonal cancers
6	Mestranol, a pharmaceutical estrogen	40 nM, MCF-7	An agonist of estrogen receptors 1 μM corresponded to 140 ± 9 treated cells for estrogenic activity, on par with 10 nM daidzein:125 ± 7 cells Exhibited estrogen binding efficacy on par with TAM and EE	Potent sideeffects include nausea, breast tension, edema, and breakthrough bleeding Initial semester usage may aggravate the risk of blood clotting Must not be taken by smokers as it aggravates the risks further No in vivo screening leaves the findings infeasible for clinical screening
7	Coumestrol, a phytoestrogen	40 nM, MCF-7 Almost water insoluble	100 nM extent caused a 100% estrogen binding efficacy over the daidzein’s 10 nM and 1 μM genistein Exhibited a weaker estrogen binding (over most pharmaceuticals) with a weakly impaired proliferation	Analysis in only MCF-7 cells leaves the results in jeopardy for TNBC cells (ER negativity)
8	Diethylstilbestrol, a synthetic non-steroidal estrogen	0.4 nM, MCF-7 BC cells Poor water solubility of 0.012 g·L^−1^ at 25 °C	Has a higher oral bioavailability than E2,more resistant to metabolism Proximal to ERs in female reproductive tract, and mammary glands Potent effects in liver and uterus Highest cell number (350 ± 23) as ER engaged compared to 10 nM genistein and daidzein (each) and 1 nM quercetin and luteolin (each) Higher than EE (10 nM) and TAM (100 nM) treated cells as ER bound	Used amounts are very high than estrogenic extents of ~2.5 nM (in various phases of menstrual cycle) No in vivo screening leaves the findings infeasible for clinical screening Analysis in only MCF-7 cells leaves the results in jeopardy for TNBC cells (ER negativity)
9	Naringenin, a phytoestrogen	0.3 μM, MCF-7 0.5 g·L^−1^ water solubility at 20 °C	Binds on par with estrogen as that of genistein, daidzein, luteolin, coumestrol Stronger ERβ proximity Weaker estrogen binding over the pharmaceutical estrogens Concentration driven estrogen agonist and antagonist behavior Anti-estrogenic effects at higher extents	Used extents are much higher than estrogenic extents of ~2.5 nM (in various phases of menstrual cycle) No in vivo screening leaves the findings infeasible for clinical screening Analysis in only MCF-7 cells leaves the results in jeopardy for TNBC cells (ER negativity)
10	Kaempferol, a phytoestrogens (173)	50 μM, MCF-7 Low water solubility	Alongwith quercetin, exhibited very weak estrogen binding activity No inhibition of MCF-7 BC proliferation by excess concentrations of pharmaceutical estrogens or of industrial wastes: xenoestrogens Maximum estrogenic activity at 10 μM	Used amounts are very high than estrogenic extents of ~2.5 nM (in various phases of menstrual cycle) No in vivo screening leaves the findings infeasible for clinical screening Analysis in only MCF-7 cells leaves the results in jeopardy for TNBC cells (ER negativity)
11	Chrysin, a phytoestrogen	50 nM, MCF-7 Almost water insoluble	Exhibits estrogenic as well as anti-estrogenic actions Binds with ERα, ERβ and E2 with same binding energies Highest estrogen binding amongst the screened flavones Maximum estrogen binding was attained at 100 μM (50% enhancement) Genistein and coumestrol equivalent estrogen binding abilities	Dual nature leads to complex interactive events in the estrogen signaling Concerns toward estrogen blocking ability Aggravates the risk of bleeding during or after surgery Exhibits toxic outcomes at higher extents
12	Apigenin, a phytoestrogen	1 μM, MCF-7 1.35 μg·mL^−1^ water solubility	Decreased the MCF-7 BC cell growth Exhibited least estrogen binding activity (at 1 μM) amongst studied flavones Maximum estrogen binding at 10 μM (leading to 160% increment) Exhibits estrogenic as well as anti-estrogenic activity, stimulates BC growth at low concentrations	Used extents are much higher than estrogenic extents of ~2.5 nM (in various phases of menstrual cycle) No in vivo screening leaves the findings infeasible for clinical screening Analysis in only MCF-7 cells leaves the results inconclusive for TNBC cells (ER negativity)
13	Luteolin, a phytoestrogens (173)	5 μM, MCF-7 Water solubility of 0.0064 mg·mL^−1^	Decreased the MCF-7 BC cell growth Exhibited more than apigenin but less than chrysin estrogenic activity (0.5 μM) Showed maximum estrogenic activity at least extent amongst the studied flavones (1 nM at 120% increment) Caused a stronger inhibition of pharmaceutical estrogen sources over those of industrial chemicals Estrogenic activities are weaker than for ES and DES	Used extents are much higher than estrogenic extents of ~2.5 nM (in various phases of menstrual cycle) No in vivo screening leaves the findings infeasible for clinical screening Analysis in only MCF-7 cells leaves the results in jeopardy for TNBC cells (ER negativity)
14	Genkwanin, a phytoestrogen	1.6 μM for MDA-MB-468 BC and 75 μM MCF-10A (normal) cells <1 μg·mL^−1^ water solubility	Binds estrogen with a potent proximity toward ERβ Estrogen binding affinity is substantially lower than 17β-estradiol (E2) Exhibits subtype based effects on BC growth but inhibits for MCF-7 cells Can inhibit estrogen functions by restraining DNA topoisomerases and tyrosine kinases	Effects vary for individuals based on health status, genetic background and the dosage texture Conclusive evidence via in vivo studies is needed to assess genkwanin safety Efficacy for cancer treatment, weak bones and other hormone related conditions requires the scrutiny via conclusive clinical trials

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
