# Peer review of "Exogenous Estrogens as Breast Cancer Risk Factors: A Perspective"

_cancers, 2025, doi:10.3390/cancers17162680_

Round 1

Reviewer 1 Report

Comments and Suggestions for Authors

This study revise estrogens receptors in breast cancer and relationship with risks of exogenous estrogens. It is very detailed review. However, what is the risk/benefit of these chemical compounds in breast cancer? Is there any final conclusion? Physiological hormonal aspects could be discussed deeper to atract the interest of medical doctors, figures with the mechanisms of action would improve the comprehension.

(1) Why Figure 3 orientation of the chemical compounds is oblique?

(2) Section 1. Please add tables and figures that summirize the most important information and function of estrogens.

(3) What is the purpose of showing/highligting the hydrophobic and hydorphilic tails? why it is relevant for the reader to know that OH is strongly polar, and is thus very hydrophilic?

(4) Section 2.2. A meta-analysis is referenced. Could you please show the Hazard-risks, forest plots, etc.?

(5) Figure 5 shows some xenoestrogens. Is the docking mechanims with estrogen receptors the same? Is it feasible to show 3D images?

(6) Section. 3.2. Regarding DDT. It is stated "considered a potential environmental carcinogen by the International Agency for Research on Cancer (IARC)". Please confirm it is only "potential".

(7) Regarding section 3.4., about the polycyclic aromatic hydrocarbons, please mention how there are present/generated during culinary/diet habits. For example, curation of meat, smoking, barbecue, soups, etc.

(8) Section 3.5. How are dioxins generated? Why are dioxins toxic? Are dioxins xenoestrogens?

(9) Table 1. Please describe with more details the effects of curcumin.

(10) Does curry powder contain phytoestrogens?

(11) Images of figure 8 are flatened. Please make sure all chemical componds are shown with same sizes, orientations, format, etc to keep consistence in the paper.

(12) In conclusions. After looking into this review. What is the conclusion regarding the risk of estrogens for breast cancer? Should perimenopause women use them or not? Are the risk/benefits properly analyzed? Should women be concerned about environmental exogenous estrogens? Could you please  make a cochrane systematic review-style analysis?

(13) Apart from breast cancer. Other neoplasia respond to estrogens. Please deep into this from a clinical point of view.

Author Response

REVIEWER 1: COMMENTS AND SUGGESTIONS FOR THE AUTHOR

GENERAL COMMENTS OF REVIEWER 1

This study revises estrogens receptors in breast cancer and relationship with risks of exogenous estrogens. It is very detailed review. However, what is the risk/benefit of these chemical compounds in breast cancer? Is there any final conclusion? Physiological hormonal aspects could be discussed deeper to attract the interest of medical doctors, figures with the mechanisms of action would improve the comprehension.

OUR RESPONSE

The introduction is revised. We have discussed the physiological roles of estrogens in both pre- and post-menopausal women, besides adding one figure (Fig.1) and one table (Table 1) mentioning the important biological functions (both shown below) of estrogen.

Functional specification

In Females

In Males

Reproductive

·   Regulation of events affecting sexual and reproductive health

·   Rising extents during puberty assist the formation of secondary sexual organs, such as breasts and whole body variations

·   Plays a key role in menstrual cycle: regulating simultaneous FSH, LH and progesterone prevalence-for regular periods

·   Regulates ovulation alongside thickening the uterus lining for preparation of pregnancy

·   Assists fertilization by thinning the cervical mucus-facilitating pregnancy by keeping vaginal walls thick, elastic and lubricated

·   Impacts sexual functions such as the erection and sperm generating ability

·   Deficiency impairs reproductive functions similar to its excess driven infertility and erectile inactions

·   Excessive estrogen is risky too, causing enlarged breasts

·   Assists in formation of testes and the sexual desire-regulating progesterone secretion from pituitary gland

·   Regulate the fluid reabsorption from the efferent ducts, maintaining optimal prostate conditions

Non-reproductive

·  Sustains important events in skeletal, cardiovascular and central nervous systems

·  Regulates blood sugar level alongwith bone and muscle mass

·  Maintains blood circulation and collagen formation for skin health

·  Low estrogen levels result in weakened bones-making them vulnerable to fracture

·   Improper estrogen levels attribute to endocrine and gastrointestinal disorders

·  Involved in neuro-protection (brain functioning), DNA repair, emotional and stress responses

·  Regulate skin elasticity, fat distribution (conferring typical body morphology)

·  Estradiol deficiency impairs the cognitive events, such as spatial ability and verbal memory

·  Impact the communication of neurons, regulating interconnections-crucial for learning and memory

·  Estrogen deficiency can result in bone loss and increased fracture risk

·  Influence mood and skin health

·  Vasodilatiry and guarding impact on cardiovascular system

·  Regulation of body weight (cholesterol metabolism) and growth of bodily hair

·  Healthy functioning of blood vessels, fluid re-absorption and cognitive coordination

We have also added the statement that the US FDA has approved the therapeutic use of pharmaceutical estrogens against various pathophysiological/diseased conditions. Also, we have mentioned the risk of using estrogens or exposure to xenoestrogens for the possible cancer development of various organs. The following line is added in the conclusion of the manuscript: When using pharmaceutical estrogens for therapeutic purposes in premenopausal women, caution is necessary, and consulting a physician is strongly recommended, especially regarding dose, duration, and type of estrogen administered”. We have briefly mentioned the risks or benefits of using plant food-derived phytoestrogens.

SPECIFIC COMMENTS OF REVIEWER 1

REVIEWER 1 COMMENT 1: Why Figure 3 orientation of the chemical compounds is oblique?

OUR RESPONSE

Figure 3 depicts the chemical structures of ethinyl estradiol (EE) and mestranol and distinguishes them from estradiol (E2). No structure is deliberately shown with an oblique orientation; the placement of functional groups is at the opposite ends of the central plane. Apart from this, the labeling of select structural features might have resulted in this confusion. In the revised manuscript, we have tried to improve the labeling, which could probably resolve this issue. The structures are depicted as generated from ChemDraw.

REVIEWER 1 COMMENT 2: Section 1. Please add tables and figures that summarize the most important information and function of estrogens.

OUR RESPONSE:  In the revised manuscript, one figure (Fig.1) and one table (Table 1) are added in the introduction describing various physiological roles of estrogens. Both are discussed above in the response to General Comments.

REVIEWER 1 COMMENT 3: What is the purpose of showing/highligting the hydrophobic and hydorphilic tails? Why it is relevant for the reader to know that OH is strongly polar, and is thus very hydrophilic?

OUR RESPONSE

We have added a separate new section on page 3 of the manuscript after the introduction with the following heading,

Section 2:“Structure-function relationship of various natural and synthetic estrogens”.

The important points of this section 2 are as follows:

  • The phenolic hydroxyl group present at the C3 position of the A ring of 17β-estradiol (the major physiological estrogen in female circulation) can be oxidized either by accepting an electron or by losing a proton (Kumar S, 2010). Thus, under some conditions, estrogens might act as a pro-oxidant, leading to the complications of diseased conditions, prominently cancers. As the structure was analysed, estrone (E1), estradiol (E2), estriol (E3), and estetrol (E4) possess one, two, three, and four hydroxyl groups, respectively.

New Reference Added:

Kumar S, Lata K, Mukhopadhyay S and Mukherjee TK. (2010) Role of estrogen receptors in pro-oxidative and anti-oxidative actions of estrogens: A perspective. BBA. 1800:1127-1135.

  • As compared to 17β-estradiol, 2-OH-E2 binds ER with a lower affinity, decreasing its BC causing potency. Both 2-OH-E2 and 4-OH-E2 reveal carcinogenic effects. However, 4-OH-E2 exhibits overproduction in BC. Additionally, 4-OH-E2 is capable of binding to ER at a low dissociation rate. Therefore, prolonged activation of the ERs induces uncontrolled cell proliferation (Cheng, Z. N, 2001).

New Reference Added:

Cheng, Z. N.; Shu, Y.; Liu, Z. Q.; Wang, L. S.; Ou-Yang, D. S.; Zhou, H. H. (2001). Role of cytochrome p450 in estradiol metabolism in vitro. Acta Pharmacologica Sinica. 22 (2): 148–154.

  • The ethinyl group at the 17-position of EE significantly enhances its estrogenic potential. The presence of the ethinyl group increases the oral bioavailability of EE when compared with 17β-estradiol. A noted effort by Helgason S, et al measured the level of pregnancy zone protein/pregnancy-associated alpha 2-globulin (PZP/PAα2 G) in postmenopausal women and suggested EE as 500 times more potent than 17-β estradiol and 650 times more potent than estradiol valerate (Helgason S, 1982).

New Reference Added

Helgason S, Damber MG, von Schoultz B & Stigbrand T. 1982. Estrogenic potency of oral replacement therapy estimated by the induction of pregnancy zone protein. Acta Obstetrica Gynecologica Scandinavica 61 75-79.

REVIEWER 1 COMMENT 4: Section 2.2. A meta-analysis is referenced. Could you please show the Hazard-risks etc.?

OUR RESPONSE

Section 2.2 is now labelled as section 3.2. The following lines are added in this section:

In this study the results of the meta-analysis mentioned that women who were taking combined oral contraceptives after 10 years of termination, there is a small increase in the relative risk of having BC diagnosed (relative risk [95% Cl]) in current users 1·24; 1-4 years after stopping 1·16, 5-9 years after stopping 1·07. The meta-analysis further mentioned that there is no significant excess risk of having BC diagnosed 10 or more years after terminating the contraceptives use (relative risk 1·01). 

REVIEWER 1 COMMENT 5: Figure 5 shows some xenoestrogens. Is the docking mechanisms with estrogen receptors the same? Is it feasible to show 3D images?

OUR RESPONSE

The following lines have been added in section 4 (previously labelled section 3)

A comparative docking study of estrogens and xenoestrogens with ERs infers that both estrogens and xenoestrogens are capable of binding the ERs, although with distinct affinities. Most of the xenoestrogens bind the ERs with a low affinity. When bound with xenoestrogens, the ERs show a distinct conformation alteration over those of estrogen binding to ERs. Several studies developed molecular modeling methods to study comparative docking between natural estrogens and xenoestrogens (Mazurek AH, 2020; Amadasi A, 2009).

We do not have the requisite permission to use any of the published docking figures from other authors. However, the references mentioned show several figures on the related topic.

New Reference Added:

  1. Mazurek AH, Szeleszczuk L, Simonson T, Pisklak DM. Application of Various Molecular Modelling Methods in the Study of Estrogens and Xenoestrogens. Int. J. Mol. Sci. 2020; 21, 6411; doi:10.3390/ijms21176411
  2. Amadasi A, Mozzarelli A, Meda C, Magg A, Cozzini P. Identification of xenoestrogens in food additives by an integrated in silico and in vitro approach. Chem Res Toxicol. 2009; 22(1): 52-63.

REVIEWER 1 COMMENT 6:  Section. 3.2. Regarding DDT. It is stated "considered a potential environmental carcinogen by the International Agency for Research on Cancer (IARC)". Please confirm it is only "potential".

OUR RESPONSE:

Section 3.2 is now labelled as section 4.2 with the addition of the following line:

In 1991, the International Agency for Research on Cancer (IARC) classified DDT as a group 2B compound and rated it possibly related to cancer (IARC, 1991).

New Reference Added

IARC (International Agency for Research on Cancer) DDT and associated compounds. IARC Monogr Eval Carcinog Risk Hum. 1991; 53:179-249

REVIEWER 1 COMMENT 7: Regarding section 3.4., about the polycyclic aromatic hydrocarbons, please mention how there are present/generated during culinary/diet habits. For example, curation of meat, smoking, barbecue, soups, etc.

OUR RESPONSE

Section 3.4 is now labelled as section 4.4 with the addition of the following lines:

A major source of PAHs is the incomplete combustion of fossil fuels [78]. The polycyclic aromatic hydrocarbons (PAH) are generated by the processing of cooking food materials (e.g., fats and proteins), such as barbecuing/grilling, roasting, and smoking. The high temperature generated during these cooking procedures causes incomplete combustion or pyrolysis, producing these hydrocarbons (Phillips DH, 1999).

New Reference Added

Phillips DH, 1999. Polycyclic aromatic hydrocarbons in the diet. Mutation Research/Genetic toxicology and environmental mutagenesis. 1999 (443) 139-147.

REVIEWER 1 COMMENT 8: Section 3.5. How are dioxins generated? Why are dioxins toxic? Are dioxins xenoestrogens?

OUR RESPONSE

Section 3.5 is now labelled as section 4.5. Based on the reviewer’s comment, we have modified section 4.5 as follows:

4.5. 2,3,7,8-. Tetrachlorodibenzo-p-Dioxin

2,3,7,8-tetrachlorodibenzo-p-Dioxin (TCDD) originates from the combustion or emissions of electric power plants, iron and steel industries, and other metal manufacturing facilities [85]. This lipophilic molecule accumulates in fat tissues, contributing to electrolyte imbalances and impaired renal function.The high doses of dioxin are related to cardiac arrhythmias, gastrointestinal issues, and neurological symptoms, and therefore, dioxin is considered highly toxic (Mathew N, 2025). Several investigations show that dioxins binding to the aryl hydrocarbon receptor (AhR) interacts with ERs, affecting multiple estrogen signaling pathways. TCDD, in particular, is known to disrupt the endocrine system, which is responsible for hormone generation and regulation. Thus, dioxin is considered an environmental xenoestrogen and shows carcinogenic activities, including being a BC risk factor [86,87]. The in vitro attempts also confirmed that TCDD promotes BC via suppression of apoptosis and anchorage-independent growth of the human breast epithelial cells with stem cell characteristics [88, 89]. The findings were reciprocated in the rodent uterus, wherein TCDD exposure enhanced the uteri’s susceptibility to BC development [90]. Though studies have revealed TCDD tumor-promoting actions through hemopoietic neoplasms, lymphatic disorders, brain and biliary complications, no such association has yet been demonstrated for BC risk [91,92].

New Reference Added

Nikhila Mathew,   Arvindh Somanathan,   Abha Tirpude,   Anupama M. Pillai,   Pabitra Mondal and  Tanvir Arfin, Dioxins and their impact: a review of toxicity, persistence, and novel remediation strategies. Anal. Methods 2025, 17, 1698-1748. DOI: 10.1039/D4AY01767.

REVIEWER 1 COMMENT 9:

Table 1. Please describe in more details the effects of curcumin.

OUR RESPONSE

Table 1 is now labelled as Table 2. In the text (section 5), the following lines are added:

One of the important phytoestrogens with potential anticancer activities is curcumin, although curcumin shows no structural relationship with estrogens or strong affinity towards ERs. Still, a few studies also indicated the ability of curcumin to influence the ERα translocation from the cytoplasm to the nucleus and its transactivation (Liu H, 2018). Several limitations are identified for the successful use of curcumin as an anticancer agent, including its capacity to behave both as an antioxidant and pro-oxidant, under certain conditions (Malik P, 2020; Malik P, 2014).

New Reference Added

  1. Haixin Liu, Shuang He, Taiyi Wang, Barnabas Orang-Ojong, Qing Lu, Zhongqun Zhang, Lanlan Pan, Xin Chai, Honghua Wu, Guanwei Fan , Peng Zhang, Yuxin Feng, Yun Seon Song, Xuimei Gao, Richard H Karas, Yan Zhu. Selected phytoestrogens distinguish roles of ERαtransactivation and ligand binding for anti-inflammatory activity free. 159 (2018), 9, 3351-3364, DOI: 10.1210/en.2018-00275.
  2. Malik P, Maktedar SS, Avashthi G, Mukherjee TK, Singh M. Robust curcumin-mustard oil emulsions for pro to antioxidant modulation of graphene oxide. Ar. J. Chem. 2020; 13(3):4606-4628.
  3. Malik P and Mukherjee TK. (2014) Structure-function elucidation of antioxidative andprooxidative activities of polyphenolic compound curcumin. Article ID 396708. Chinese J. Biol. 17:286-291.

REVIEWER 1 COMMENT10: Does curry powder contain phytoestrogens?

OUR RESPONSE

Yes, curry powder contains several molecules, such as turmeric, which are the potential source of phytoestrogens.

The following line is added to the manuscript in section 5:

People of the Indian subcontinent and Southeast Asian countries use various spices, including curry powder, for cooking, which contains several phytoestrogens.

REVIEWER 1 COMMENT 11: Images of figure 8 are flattened. Please make sure all chemical compounds are shown with same sizes, orientations, format, etc to keep consistence in the paper.

OUR RESPONSE

In the revised manuscript, Fig.8 has been thoroughly re-checked, and all the depicted compounds are now in similar font sizes and format. Figure has been saved again from ChemDraw Image option, with the best attempts to avoid flattening.

REVIEWER 1 COMMENT 12: In conclusions. After looking into this review. What is the conclusion regarding the risk of estrogens for breast cancer? Should peri-menopause women use them or not? Are the risk/benefits properly analyzed? Should women be concerned about environmental exogenous estrogens? Could you please make a cochrane systematic review-style analysis?

OUR RESPONSE

Based on your worthy comments, we have added a completely new conclusion.

Here are the details:

  1. Conclusions and future perspectives

BC is the most common cancer worldwide and the second leading cause of cancer-related deaths. In premenopausal women, although the ovaries are the major sources of estrogens, adipose tissue and adrenal glands contribute small amounts of estrogens. Menopause significantly decreases estrogen production from the ovaries. However, adipose tissue, adrenal glands, and tissues such as the brain and lungs continue to produce estrogens due to the expression of the aromatase enzyme at the local tissue level. Besides natural estrogens, the human body, including both pre- and postmenopausal women, regardless of age and menstrual status, is exposed to various pharmaceutical estrogens, environmental xenoestrogens, and food-derived phytoestrogens. Many of these compounds, whether pharmaceutical estrogens, xenoestrogens, or phytoestrogens, bind to ERs with different affinities and cause diverse actions. At the physiological level, the interaction between estrogen and ERs regulates reproductive functions as well as other non-reproductive processes. Under certain conditions, natural, synthetic, or environmental estrogens can trigger neoplastic transformations in human epithelial cells, leading to cancers of the breast, ovaries, uterine endometrium, and lungs.

Therefore, when using pharmaceutical estrogens for therapeutic purposes in women, caution is necessary, and consulting a physician is strongly recommended, especially regarding dose, duration, and type of estrogen administered. Physicians' recommendations and advice are highly essential regarding the dose of hormonal therapy (low dose versus high dose) and whether monotherapy (estrogen-only therapy) or combination therapy (estrogen + progesterone) may be used. Premenopausal women are particularly vulnerable to the effects of pharmaceutical estrogens due to the high physiological levels of estrogen produced by their ovaries. In postmenopausal women, endogenous estrogen synthesis sharply decreases, leading to symptoms such as osteoporosis, vaginal dryness, and increased risks of CVDs and other conditions. Several studies, including the Nurses' Health Study (NHS), Women’s Health Initiative (WHI), and others, show the benefits of hormone replacement therapy (HRT) in alleviating postmenopausal symptoms. However, other research contradicts these benefits and warns against continued estrogen use because of potential cancer risks and adverse effects on the cardiovascular system. Besides natural and synthetic estrogens, the human body is exposed to environmental xenoestrogens from sources such as pesticides, plastic containers, and cosmetics, which can disrupt the physiology of endogenous estrogens and thus are regarded as endocrine-disrupting chemicals (EDCs). Precautions should be taken to minimize exposure to xenoestrogens due to their ability to cause genetic and epigenetic modifications (particularly DNA methylation and acetylation) and promote the development and progression of various cancers. Many studies suggest that pro-carcinogenic xenoestrogens tend to bind strongly to ERα, promoting cancer development.

In contrast, phytoestrogens, a group of diverse compounds found in foods such as soy products, may offer health benefits or even protection against the development of cancers. Diverse mechanisms of protective actions of phytoestrogens are observed, including antioxidative and anti-inflammatory actions. Additionally, a comparative study by Gustafson and his team (the discoverer of ERβ) showed that 17β-estradiol, the most abundant natural estrogen in women, has nearly equal affinity for ERα and ERβ. On the other hand, phytoestrogens like genistein, derived from soy, exhibit ~400 times greater affinity for ERβ, which contributes to their protective effects against cancers and CVDs. Other studies noted that the protective effects of phytoestrogens on human health, including their role in preventing BCs, depend on factors such as ethnicity (e.g., South-Southeast Asian people), although the majority of the study results claimed that it is the specific food habits, rather than genetics which takes a decisive role of the protective actions of phytoestorgens. Although several clinical studies have explored the potential benefits of phytoestrogens, conclusive evidence confirming their role in cancer prevention is still lacking. The patient-derived xenograft models and nanoparticle-dependent drug delivery are important areas of present-day research in breast cancer therapy (Li J, 2025; Ma P, 2024; Huang M, 2023). More research is needed on the impact of environmental estrogens/phytoestrogens, including the effects of combination exposure, keeping in mind the sex, age, and other critical factors, which would further enhance our knowledge in this area.

REVIEWER 1 COMMENT 13: Apart from breast cancer. Other neoplasia respond to estrogens. Please deep into this from a clinical point of view.

OUR RESPONSE

The following paragraph is added in the introduction:

Natural estrogens and some of their endogenous metabolites, such as 2-OH-E2 and 4-OH-E2, can induce mutations in epithelial cells and exhibit carcinogenic effects under certain conditions [5]. In addition to genetic alterations (e.g., BRCA1 and BRCA2 mutations), epigenetic changes and various environmental factors influence estrogen’s physiological and pathological actions. Estrogens, their metabolites, and androgens not only contribute to the development of breast cancers but also are linked to ovarian, uterine endometrial cancers in women, prostate cancer in men (Miki Y, 2020; Ulm M., 2019), and lung cancers in both sexes (16. Maitra R, 2021). Furthermore, estrogen receptors (ERs) and estrogen-related receptors (ERRs, a group of orphan nuclear receptors related to estrogen actions) also play roles in cancer development (16. Maitra R, 2021). Epidemiological, clinical, and experimental studies suggest that besides natural and synthetic estrogens, long-term exposure to high levels of xenoestrogens may promote the growth and progression of breast, ovarian, and endometrial cancers [11,13, 14]. Regarding their genomic actions, estrogens bind to their receptors, translocate to the nucleus, recognize and bind to estrogen-responsive elements (ERE), activate transcription of genes involved in cancer development and propagation (Bjornstrom L, 2005). However, 10-15% of BC cases lack ERs, PRs, and HER2 (TNBCs), meaning estrogens are not involved in the development or complications of TNBCs (Kumar P, 2016). Clinically, while estrogen-synthesizing aromatase enzymes and ERs are targeted in estrogen-ER-dependent cancers, other therapeutic strategies such as chemotherapy, targeted therapy, and immunotherapy are used to treat TNBCs (Leon-Ferre RA, 2023). This comprehensive review explores the role of various exogenous estrogens, including pharmaceutical estrogens, xenoestrogens, and phytoestrogens, as potential risk factors for BC.

  New Reference Added:

  1. Miki Y. New Insights into breast and endometrial cancers. Cancers 2020;12:2595. DOI: 10.3390/cancers12092595. 
  2. Ulm M., Ramesh A.V., McNamara K.M., Ponnusamy S., Sasano H., Narayanan R. Therapeutic advances in hormone-dependent cancers: Focus on prostate, breast and ovarian cancers. Endocr. Connect. 2019;8:R10-R26.
  3. Kumar P, Aggarwal R. An overview of triple negative breast cancer. Arch Gynecol Obstet. 2016; 293(2):247-69.
  4. Bjornstrom L, Sjoberg M. Mechanism of estrogen receptor signaling: Convergence of genomic and nongenomic actions of target genes. Endocrinol. 19 (2005) 833-842.
  5. Leon-Ferre RA, Goetz MP. Advances in systemic therapies for triple negative breast cancer. 2023; 381:e071674.

Reviewer 2 Report

Comments and Suggestions for Authors

This article primarily investigates the correlation between high-level exposure to endogenous estrogens and their metabolites (including estradiol, estriol, 2-hydroxyestradiol, and 4-hydroxyestradiol) and the onset and complications of breast cancer (BC). Beyond endogenous estrogens, humans encounter estrogen and estrogen-like substances through various environmental sources, encompassing medicinal estrogens, xenoestrogens, and phytoestrogens. For instance, women consume medicinal estrogens during postmenopausal hormone therapy (HRT) and through oral contraceptives, either independently or in conjunction with progestins. Furthermore, individuals (inclusive of women) are exposed to estrogen-like chemicals, ubiquitously found in pesticides, plastics, and personal care items, via inhalation, dermal contact, and ingestion. Additionally, numerous phytoestrogens (such as isoflavones and lignans) enter the human body as dietary components. The author is advised to revise the manuscript in accordance with the following comments.

  1. It is recommended that the author allocate additional space in the introduction to present comprehensive background information on breast cancer.
  2. The following references are closely related to the author's research topic and are recommended for citation.

[1] J. Li, A. Gu, N. Tang, G. Zengin, M.-Y. Li, Y. Liu, Patient-derived xenograft models in pan-cancer: From bench to clinic. Interdiscip. Med. 2025, 3, e20250016. DOI: 10.1002/INMD.20250016

[2] Huang M, Gong G, Deng Y, Long X, Long W, Liu Q, et al. Crosstalk between cancer cells and the nervous system. Med Adv. 2023; 1(3): 173–189. https://doi.org/10.1002/med4.27

[3] P. Ma, G. Wang, K. Men, et al. Advances in clinical application of nanoparticle-based therapy for cancer treatment: A systematic review, Nano TransMed. 3 (2024) 100036. https://doi.org/10.1016/j.ntm.2024.100036

  1. The author is encouraged to conduct an in-depth examination of the molecular mechanisms through which exogenous estrogen stimulates breast cancer development. This could include, but is not limited to, the signal transduction of estrogen receptors and gene regulatory networks, in order to enhance our comprehension of the subject matter.
  2. The discrepancies observed between experimental results suggest the need for a more comprehensive analysis by the author. Consideration of factors such as variations in experimental conditions and differences in research subjects could strengthen the reliability of the conclusions drawn.
  3. The author is recommended to gather and present detailed information concerning the exposure pathways and concentration levels of environmental estrogens, along with their correlation to breast cancer risk.
  4. The author is encouraged to investigate the potential protective mechanisms of plant estrogens, including their binding patterns with estrogen receptors and their influence on cellular signal transduction, in order to elucidate their role in the prevention of breast cancer.

Author Response

REVIEWER 2: COMMENTS AND SUGGESTIONS FOR AUTHORS

GENERAL COMMENTS OF REVIEWER 2

This article primarily investigates the correlation between high-level exposure to endogenous estrogens and their metabolites (including estradiol, estriol, 2-hydroxyestradiol, and 4-hydroxyestradiol) and the onset and complications of breast cancer (BC). Beyond endogenous estrogens, humans encounter estrogen and estrogen-like substances through various environmental sources, encompassing medicinal estrogens, xenoestrogens, and phytoestrogens. For instance, women consume medicinal estrogens during postmenopausal hormone therapy (HRT) and through oral contraceptives, either independently or in conjunction with progestins. Furthermore, individuals (inclusive of women) are exposed to estrogen-like chemicals, ubiquitously found in pesticides, plastics, and personal care items, via inhalation, dermal contact, and ingestion. Additionally, numerous phytoestrogens (such as isoflavones and lignans) enter the human body as dietary components.

OUR RESPONSE

Thanks for the analysis of the manuscript

SPECIFIC COMMENTS OF REVIEWER 2

The author is advised to revise the manuscript in accordance with the following comments.

REVIEWER 2 COMMENT 1: It is recommended that the author allocate additional space in the introduction to present comprehensive background information on breast cancer.

OUR RESPONSE

Dear Sir, This concern has been addressed in our response to Comment number 13 from Reviewer 1 We have added a separate paragraph in the introduction-describing the role of estrogens in breast cancer.

REVIEWER 2 COMMENT 2: The following references are closely related to the author's research topic and are recommended for citation.

[1] J. Li, A. Gu, N. Tang, G. Zengin, M.-Y. Li, Y. Liu, Patient-derived xenograft models in pan-cancer: From bench to clinic. Interdiscip. Med. 2025, 3, e20250016. DOI: 10.1002/INMD.20250016

[2] Huang M, Gong G, Deng Y, Long X, Long W, Liu Q, et al. Crosstalk between cancer cells and the nervous system. Med Adv. 2023; 1(3): 173-189. https://doi.org/10.1002/med4.27

[3] P. Ma, G. Wang, K. Men, et al. Advances in clinical application of nanoparticle-based therapy for cancer treatment: A systematic review. Nano TransMed. 3 (2024) 100036. https://doi.org/10.1016/j.ntm.2024.100036

OUR RESPONSE

We have added the following lines in the conclusion to include the references:

The patient-derived xenograft models and nanoparticle-dependent drug delivery are important areas of present-day research in breast cancer (Li J, 2025; Ma P, 2024; Huang M, 2023) at number 174 to 176. More research is needed on the impact of environmental estrogens as a risk factor for the development and complications of breast cancer.

REVIEWER 2 COMMENT 3: The author is encouraged to conduct an in-depth examination of the molecular mechanisms through which exogenous estrogen stimulates breast cancer development. This could include, but is not limited to, the signal transduction of estrogen receptors and gene regulatory networks, in order to enhance our comprehension of the subject matter.

OUR RESPONSE

Very brief outline is added in the last paragraph of the introduction to discuss the genomic and non-genomic actions of estrogens.

Additionally, a small separate new section (labeled as 4.6) with the following heading and material:

4.6. Mechanism of action of exogenous xenoestrogens in the development of cancers      

REVIEWER 2 COMMENT 4: The discrepancies observed between experimental results suggest the need for a more comprehensive analysis by the author. Consideration of factors such as variations in experimental conditions and differences in research subjects could strengthen the reliability of the conclusions drawn.

OUR RESPONSE

In the xenoestrogen section, we have added various molecules such as bisphenols, DDT, Polychlorinated Biphenyls, Polycyclic Aromatic Hydrocarbons, and 2,3,7,8-. Tetrachlorodibenzo-p-Dioxin discussed its mechanism of action.

In the phytoestrogen section, we have discussed various factors that influence their mechanism of action, with the following subheadings:

  • Role of estrogen receptors in phytoestrogen-dependent breast cancer complications
  • Role of menopausal status in the effects of phytoestrogen-dependent breast cancer complications
  • Concentration-dependent response of phytoestrogens on breast cancer complications.

So, no new information added in this section.

REVIEWER 2 COMMENT 5: The author is recommended to gather and present detailed information concerning the exposure pathways and concentration levels of environmental estrogens, along with their correlation to breast cancer risk.

OUR RESPONSE

The following lines added section 4 of the manuscript:

Xenoestrogens are a group of external chemical compounds created through human-made processes and industrial activities. They enter the human body through inhalation, skin contact, and the ingestion of contaminated water, food, or supplements. Major sources of xenoestrogens include personal care products, plastics, industrial byproducts, pesticides, herbicides, and by-products from cooking under high heat. The main exposure routes are dermal absorption (from personal care products and industrial environments), inhalation (airborne particles and volatile organic compounds), ingestion (food and water), and vertical transfer through the placenta and breast milk (Chmielewski J, 2021). These diverse compounds are broadly categorized as synthetic estrogens (e.g., d or DES), alkylphenols (e.g., 4-nonylphenol), phenyl derivatives (e.g., bisphenol A or BPA), and organochlorines (e.g., DDT, PCBs, parabens like 2-ethylhexyl 4-hydroxybenzoate), all exhibiting different levels of affinity for estrogen receptors.

FOR THE MECHANISM OF ACTION, THE FOLLOWING LINES ARE ADDED

4.6. Mechanism of action of exogenous xenoestrogens in the development of cancers      

The exogenous estrogens show lasting pathophysiological consequences either by directly affecting several genetic and epigenetic alterations or by competing with endogenous estrogens for ERs and disrupting the natural, physiological signaling mechanism mediated by the natural estrogen-ERs complex. Several levels of evidence indicate that both the natural physiological and pathophysiological functions of endogenous estrogens are affected by exogenous estrogens. This imbalance in estrogen levels in the body, generated by the exogenous estrogens, influences several pathophysiological events, including infertility, immunological disorders, various cancers, and several other pathophysiological consequences. Critical analysis of various cellular and molecular events associated with the development and progression of different cancers shows that exogenous estrogens affect all the cell-signalling molecules involved in cell proliferation (e.g., cyclin D), survival (e.g., AKT), apoptosis (e.g., Bcl-2), angiogenesis (e.g., VEGF/VEGFR), and invasion and metastasis (e.g., MMPs) of cancer cells. In a comprehensive review article, Wieczorek JB et al described the impact of exogenous estrogens on various cell signaling molecules and cellular events associated with physiological and pathophysiological functions (Wieczorek JB. 2024).

New References Added:

  1. Joanna Bartkowiak-Wieczorek, Agnieszka Jaros, Anna Gajdzińska, Paulina Wojtyła-Buciora, Igor Szymański, Julian Szymaniak, Wojciech Janusz, Iga Walczak, Gabriela Jonaszka, Agnieszka Bienert. The dual faces of oestrogen: The impact of exogenous oestrogen on the physiological and pathophysiological functions of tissues and organs. Int J Mol Sci. 2024; 25(15):8167. DOI: 10.3390/ijms25158167.
  2. Jarosław Chmielewski, Jarogniew Łuszczki, Małgorzata Czarny-Działak, Ewa Dutkiewicz, Halina Król, Barbara Gworek, Grażyna Nowak-Starz. Environmental exposition to xenoestrogens and related health effects. J Elem 26(3):717-730. DOI: 10.5601/jelem.2021.26.2.2157

REVIEWER 2 COMMENT 6: The author is encouraged to investigate the potential protective mechanisms of plant estrogens, including their binding patterns with estrogen receptors and their influence on cellular signal transduction, in order to elucidate their role in the prevention of breast cancer.

OUR RESPONSE

We have added a small paragraph in the phytoestrogen section

  • Possible protection mechanisms of phytoestrogen-dependent cancer complications

The effects of phytoestrogens, whether as a protector or a promoter of cancers, depend on several factors, including the type of phytoestrogen, the specific estrogen receptor it interacts with, the concentration of the phytoestrogens, and other variables. The potential anticancer activities of different phytoestrogens are generally categorized into the following protective mechanisms: 1. Modulating estrogen receptor activities, 2. Reducing pro-proliferative and pro-survival cell signaling pathways, 3. Decreasing estrogen production and altering its metabolism, 4. Epigenetic modifications, 5. Anti-oxidative and anti-inflammatory mechanisms. Several studies reported that phytoestrogens, such as genistein, have a preferential binding affinity for ERβ, which contributes to their anti-proliferative effects. In ER-negative breast cancer cell lines, phytoestrogens help prevent cancer complications through epigenetic modifications, including alteration of DNA  methylation and histone acetylation. The anti-inflammatory properties of several phytoestrogens, such as genistein, resveratrol, etc, are well established. A series of comprehensive review articles highlighted the various mechanisms through which different phytoestrogens protect against cancers, including breast cancer (Mas MT, 2020; Hsieh CJ, 2018; Bilal I, 2014; Mukherjee TK, 2003)

New References Added:

  1. Chia-Jung Hsieh, Ya-Ling Hsu, Ya-Fang Huang and Eing-Mei Tsai. Molecular mechanisms of anticancer effects of phytoestrogens in breast cancer. Curr. Protein & Peptide Sci. 19 (3), 2018 Page: [323 - 332] Pages: 10. DOI: 10.2174/1389203718666170111121255.
  2. Mukherjee TK, Nathan L, Dinh H, Reddy ST and Chaudhuri G. (2003) 17-epiestriol, an estrogen metabolite, is more potent than estradiol in inhibiting VCAM-1 mRNA expression. J. Biol. Chem. 278: 11746-11752.
  3. Iqra Bilal, Avidyuti Chowdhury, Juliet Davidson, Saffron Whitehead. Phytoestrogens and prevention of breast cancer: The contentious debate. World J Clin Oncol. 2014; 5(4):705-712. DOI: 10.5306/wjco.v5.i4.705.
  4. Margalida Torrens-Mas, Pilar Roca. Phytoestrogens for Cancer Prevention and Treatment. Biology (Basel). 2020; 9(12):427. DOI: 10.3390/biology9120427.

Reviewer 3 Report

Comments and Suggestions for Authors

The article titled “Exogenous Estrogens as Breast Cancer Risk Factors: A Perspective” has been evaluated and has been found to be lacking in organization. It is essential that the authors undertake significant revisions before the manuscript can be deemed suitable for publication in a scientific journal.

One critical area that requires attention is the articulation of the research methodology utilized in this study. The authors are advised to provide a clear and detailed description of the methodologies applied, including the study design, data collection techniques, and analytical approaches. Additionally, a flow diagram that clearly delineates the inclusion and exclusion criteria for the articles reviewed should be incorporated. This diagram must be well-labeled to enhance reader comprehension.

While the current review primarily focuses on the chemical properties of the compounds under investigation, it falls short in offering a thorough explanation of how each class of compounds interacts with biological systems. It is imperative that the authors include a comprehensive overview of the mechanisms of action for these exogenous estrogens. This should encompass relevant mechanistic pathways, ideally supplemented with an illustrative figure to visually represent these processes.

Moreover, the authors need to address the specific dosage levels at which these exogenous estrogens may pose a cancer risk. It is crucial to clearly distinguish between the therapeutic and clinical uses of these compounds, specifying the associated doses that may increase the risk of breast cancer. This distinction will provide clarity on the implications of their findings and guide future research and clinical practices.

Finally, it is strongly recommended to add a dedicated section for "Abbreviations" within the manuscript. This section should list all abbreviations used in the article along with their full definitions to enhance reader comprehension and overall readability.

Author Response

REVIEWER 3: COMMENTS AND SUGGESTIONS FOR AUTHORS

REVIEWER 3 COMMENT 1

The article titled “Exogenous Estrogens as Breast Cancer Risk Factors: A Perspective” has been evaluated and has been found to be lacking in organization. It is essential that the authors undertake significant revisions before the manuscript can be deemed suitable for publication in a scientific journal.

One critical area that requires attention is the articulation of the research methodology utilized in this study. The authors are advised to provide a clear and detailed description of the methodologies applied, including the study design, data collection techniques, and analytical approaches. Additionally, a flow diagram that clearly delineates the inclusion and exclusion criteria for the articles reviewed should be incorporated. This diagram must be well-labeled to enhance reader comprehension.

OUR RESPONSE In the revised manuscript, the selection criteria of published literature is now added and explained suitably with the help of newly added Figure 2.

REVIEWER 3 COMMENT 2

While the current review primarily focuses on the chemical properties of the compounds under investigation, it falls short in offering a thorough explanation of how each class of compounds interacts with biological systems. It is imperative that the authors include a comprehensive overview of the mechanisms of action for these exogenous estrogens. This should encompass relevant mechanistic pathways, ideally supplemented with an illustrative figure to visually represent these processes.

OUR RESPONSE

Section 4 (from 4.1 to 4.5) of this manuscript illustrates in detail the most widely studied xenoestrogenic compounds, along with their chemical structure.

However, based on the reviewer’s comment, we have added a new figure on this topic as shown overleaf. Figure 7 is on diversified sources of exogenous estrogens (sources) while Figure 9 describes the interacting targets through which estrogens enter the physiological system boundaries.

REVIEWER 3, COMMENT 3

Moreover, the authors need to address the specific dosage levels at which these exogenous estrogens may pose a cancer risk. It is crucial to clearly distinguish between the therapeutic and clinical uses of these compounds, specifying the associated doses that may increase the risk of breast cancer. This distinction will provide clarity on the implications of their findings and guide future research and clinical practices.

OUR RESPONSE

One complete paragraph is dedicated to showing the effects of various concentrations of phytoestrogens. Here is the subtitle:

  • Concentration-dependent response of phytoestrogens on breast cancer complications

The manuscript is already very large. There is no scope to further increase it.

REVIEWER 3, COMMENT 4

Finally, it is strongly recommended to add a dedicated section for "Abbreviations" within the manuscript. This section should list all abbreviations used in the article along with their full definitions to enhance reader comprehension and overall readability.

OUR RESPONSE

We have added the following abbreviations at the end of the keywords.

Abbreviations

Estrone (E1), 17β-Estradiol/estradiol(E2), Estriol (E3), and Estetrol (E4),17α-Ethinyl Estradiol (EE), 2-hydroxy-estradiol (2-OH-E2) and 4-hydroxy-estradiol (4-OH-E2),Environmental Chemical Exposures (ECE), Endocrine Disrupting Chemicals (EDCs),Bisphenol A (BPA),Dichlorodiphenyltrichloroethane (DDT),Polychlorinated Biphenyls (PCBs), Polycyclic Aromatic Hydrocarbons (PAHs), 2,3,7,8-tetrachlorodibenzo-p-Dioxin (TCDD),Breast Cancer Gene 1/2 (BRCA1/2), Estrogen Response Elements, (ERE),Estrogen receptors (ERs),Estrogen Receptor Alpha (ERα, its gene stands for ESR1), Estrogen Receptor Beta (ERβ, its gene stands for ESR2), Progesterone Receptors (PRs),  Human Epidermal Growth Factor Receptor 2 (HER-2), Estrogen Related Receptors (ERRs), Breast Cancers (BCs), Triple Negative Breast Cancers (TNBCs),Oral Contraceptive Pills (CP), Hormone Replacement Therapy (HRT),Nurses' Health Study (NHS), Women’s Health Initiative (WHI) , US Food and Drug Administration (FDA).

Reviewer 4 Report

Comments and Suggestions for Authors

The review examines various forms of exogenous estrogens but fails to elucidate the quantitative assessment of such exposures in human populations.  For instance, how do the concentrations of these compounds vary based on demographics or broad geographical regions?  Have published studies directly correlating exposure levels to incidence rates or progression of breast cancer been conducted? 

 The review paper presents a comprehensive enumeration of estrogenic compounds, failing to distinguish other confounding risk factors that may contribute to the development of breast cancer.  These may encompass genetics, lifestyle factors, and environmental exposures to substances not directly associated with estrogens.  Have studies been conducted to evaluate the concurrent interaction of these factors? 

 The review provides a concise overview of cellular and molecular mechanisms but fails to investigate the specific interactions of competing estrogenic compounds with cellular and signaling pathways associated with breast cancer.  Could the authors elaborate on the molecular interactions associated with estrogen receptors and other competing cellular mediators of breast cancer progression? 

 The review acknowledged that exposure to pharmaceutical estrogens and xenoestrogens contributes to breast cancer progression; however, it did not address the varying sensitivity of breast cancer among different types.  Could the authors elucidate how the aforementioned estrogenic compounds influenced specific breast cancer subtypes and their correlation with various treatment protocols? 

 The review article presents experimental findings; however, it could have included longitudinal studies or epidemiological norms to comprehensively elucidate causation.  Have long-term studies investigated cumulative factors, including both endogenous and exogenous exposures, to assess risk factors for breast cancer?

Author Response

REVIEWER 4: COMMENTS AND SUGGESTIONS FOR AUTHORS

REVIEWER 4, COMMENT 1

The review examines various forms of exogenous estrogens but fails to elucidate the quantitative assessment of such exposures in human populations.  For instance, how do the concentrations of these compounds vary based on demographics or broad geographical regions?  Have published studies directly correlating exposure levels to incidence rates or progression of breast cancer been conducted? 

 The review paper presents a comprehensive enumeration of estrogenic compounds, failing to distinguish other confounding risk factors that may contribute to the development of breast cancer.  These may encompass genetics, lifestyle factors, and environmental exposures to substances not directly associated with estrogens.  Have studies been conducted to evaluate the concurrent interaction of these factors? 

OUR RESPONSE

The following lines are added to section 4 of the manuscript

Various demographic factors that influence the exposure and effects of xenoestrogens on human health are genetic predisposition, age, and sex of the individual. The other demographic factors which are known to affect xenoestrogen exposure are geographic locations, lifestyle choices, dietary habits, occupational factors, and socioeconomic status. Several population-based studies have been conducted to show a positive association between serum total xenoestrogen levels and BC risk. In one such study, Barriuso et al showed the significance of EDCs-evaluating mixtures of EDCs, rather than single chemicals, in epidemiological studies on hormone-related cancers (Barriuso RP, 2016).

New Reference Added:

Roberto Pastor-Barriuso, Mariana F Fernández, Gemma Castaño-Vinyals, Denis Whelan, Beatriz Pérez-Gómez, Javier Llorca, Cristina M Villanueva, Marcela Guevara, José-Manuel Molina-Molina, Francisco Artacho-Cordón, Laura Barriuso-Lapresa, Ignasi Tusquets, Trinidad Dierssen-Sotos, Nuria Aragonés, Nicolás Olea, Manolis Kogevinas, Marina Pollán. Total effective xenoestrogen burden in serum samples and risk for breast cancer in a population-based multicase-control study in Spain. Environ Health Perspect. 2016; 124(10):1575-1582. DOI: 10.1289/EHP157. 

 REVIEWER 4 COMMENT 2

The review provides a concise overview of cellular and molecular mechanisms but fails to investigate the specific interactions of competing estrogenic compounds with cellular and signaling pathways associated with breast cancer. Could the authors elaborate on the molecular interactions associated with estrogen receptors and other competing cellular mediators of breast cancer progression? 

OUR RESPONSE

We have added the following lines in section 3 of the manuscript:

In genomic actions, estrogens bind to their receptors, forming estrogen-ER complex, which translocates to the nucleus, recognizes and binds to estrogen-responsive elements (ERE), finally activating the transcription of genes involved in cancer development and progression. While ERα is recognized as responsible for BC development, the role of ERβ is controversial (Bjornstrom L, 2005). The interaction of estrogens with ER is highly complex, and several other signaling molecules are involved in ER-dependent and ER-independent BC development and complications, including TNBCs (Kumar P, 2016). Notable signaling molecules and pathways that are involved in the complex interplay of ERs in estrogen-dependent or independent BC development and progression are growth factors, cytokines, and TGF-β1. Additionally, the extracellular matrix (ECM) components and the dynamics of interaction with the ERs emerge as a pivotal factor-modulating the epigenetic mechanisms of BC-complications (Mangani S, 2024).

New Reference Added:

Mangani S, Piperigkou Z, Koletsis NE, Ioannou P, Karamanos NK. Estrogen receptors and extracellular matrix: the critical interplay in cancer development and progression. FEBS J. 2025; 292(7):1558-1572. DOI: 10.1111/febs.17270. Epub 2024.

REVIEWER 4 COMMENT 3

The review acknowledged that exposure to pharmaceutical estrogens and xenoestrogens contributes to breast cancer progression; however, it did not address the varying sensitivity of breast cancer among different types.  Could the authors elucidate how the aforementioned estrogenic compounds influenced specific breast cancer subtypes and their correlation with various treatment protocols? 

OUR RESPONSE

The following lines are added in section 3 of the manuscript

While working through ERs, xenoestrogens can complicate BC in the absence of ERs (in TNBC cells), by interacting with receptors such as G-protein coupled estrogen receptors (GPERs), estrogen-related receptor gamma (ERRγ), and others (Gachowska M, 2024). Additionally, xenoestrogens can affect the tumor immune microenvironment (TIME). Thus, therapeutic strategies of both hormone-dependent and independent cancers are influenced by exogenous xenoestrogen concentrations and exposure durations (Fucic A, 2012).

New References Added:

  1. Gachowska M, Dąbrowska A, Wilczyński B, Kuźnicki J, Sauer N, Szlasa W, Kobierzycki C, Łapińska Z, Kulbacka J. The influence of environmental exposure to xenoestrogens on the risk of cancer development. Int J Mol Sci. 2024 Nov 18;25(22):12363. doi: 10.3390/ijms252212363.
  2. Aleksandra Fucic, Marija Gamulin, Zeljko Ferencic, Jelena Katic, Martin Krayer von Krauss, Alena Bartonova, Domenico F Merlo. Environmental exposure to xenoestrogens and oestrogen related cancers: reproductive system, breast, lung, kidney, pancreas, and brain. Environ Health. 2012 Jun 28;11 Suppl 1(Suppl 1):S8. doi: 10.1186/1476-069X-11-S1-S8.

REVIEWER 4, COMMENT 5

The review article presents experimental findings; however, it could have included longitudinal studies or epidemiological norms to comprehensively elucidate causation.  Have long-term studies investigated cumulative factors, including both endogenous and exogenous exposures, to assess risk factors for breast cancer?

OUR RESPONSE

The reviewer is right to say that longitudinal studies or epidemiological norms should have been included in the review. However, we did not find too many systematic reviews and literature surveys on the above topic. Very briefly, we have added the following lines in section 4 of the manuscript to address reviewers' concerns:

Several population-based studies have been conducted to show a positive association between serum total xenoestrogen levels and BC risk. In one such study, Barriuso and accomplices showed the importance of evaluating EDCs mixtures, rather than single chemicals, in epidemiological studies on hormone-related cancers (Barriuso RP, 2016). In 2020, Zeinomer N and team conducted a systematic review using 100 publications on epidemiological studies on environmental chemical exposures (ECE) and BC risk in women. Their results concluded that women at higher BC risk through family history, younger onset age, and genetic susceptibility, consistently supporting the association between an ECE and BC risk (Zeinomer N, 2020). In a recent meta-analysis study of 67 publications, Liu and group showed a statistically significant association of several EDCs (organochloride pesticides, PCBs, phthalates, diesters-their metabolites, PFAS, flame retardants, and BPA) as BC risk factors (Liu H, 2023).

New References Added:

Haiyan Liu, Yukun Sun, Longkai Ran, Jiuling Li, Yafei Shi, Chunguang Mu, Changfu Hao . Endocrine-disrupting chemicals and breast cancer: a meta-analysis. Front Oncol. 2023; 13:1282651. DOI: 10.3389/fonc.2023.1282651.

Nur Zeinomar, Sabine Oskar, Rebecca D Kehm, Shamin Sahebzeda , Mary Beth Terry. Environmental exposures and breast cancer risk in the context of underlying susceptibility: A systematic review of the epidemiological literature. Environ Res. 2020; 187:109346. DOI: 10.1016/j.envres.2020.109346.

ADDITIONALLY, THE FOLLOWING LINES ARE ADDED IN SECTION 3.1:

The Nurses' Health Study (NHS) and its 10-year follow-up after adjusting for age and other risk factors claimed that the overall relative risk of major coronary disease in women currently taking estrogen was 0.56 (95 percent confidence interval, 0.40 to 0.80). The results indicate that the risk of coronary diseases was significantly reduced among women with either natural or surgical menopause (Stampfer MJ, 1991). Several studies have claimed that pharmaceutical estrogens used in hormone replacement therapy (HRT) and oral contraceptive pills increase BC risk. Conversely, other studies argue that progesterone, not estrogen, in contraceptive pills is what elevates this risk. An eminent Women’s Health Initiative (WHI) study claimed that systematic estrogen monotherapy does not increase BC risk, and perhaps, in certain conditions, it may even lower the BC risk. However, systemic combination therapy increases the BC risk, even after ten years of HRT use termination in females. Additionally, higher doses of combined HRT (estrogen and progestins) confer a greater BC risk via relapse in advanced BC patients [25, 26]. In a WHI randomized trial, estrogen + progesterone therapy and family history were screened as independent (non-interacting) factors towards invasive BC risk amongst the sufferers [27].

New Reference added:

M.J. Stampfer, GA Colditz, W.C. Willett, JE Manson, B. Rosner, F.E. Speizer, C.H. Hennekens. Postmenopausal estrogen therapy and cardiovascular disease. Ten-year follow-up from the nurses' health study. N Engl J Med. 1991 Sep 12;325(11):756-62.

Reviewer 5 Report

Comments and Suggestions for Authors

This manuscript provides a narrative review of the current understanding of how exogenous estrogens, derived from pharmaceuticals, environmental contaminants, dietary sources, and other external exposures, may contribute to breast cancer risk. The topic is timely and of clinical and public health relevance. However, the review suffers from a lack of depth, limited critical synthesis of the literature, and insufficient mechanistic or epidemiological detail. The manuscript currently reads as a general overview rather than a high-impact, analytical review expected by Cancers.

  1. The manuscript lacks a clear organizing structure or conceptual framework. A more rigorous breakdown, such as by type of exposure (e.g., pharmaceutical, environmental, dietary), mechanism of action (e.g., ER-dependent vs independent), or level of evidence (e.g., in vitro, in vivo, clinical, epidemiological), would greatly improve readability and coherence.
  2. Key landmark studies and meta-analyses are either missing or superficially mentioned. For instance, large-scale epidemiological studies from WHI, EPIC, or Nurses’ Health Study, which significantly shaped current understanding of estrogen-related breast cancer risk, are not adequately reviewed or contextualized.
  3. The mechanistic insights are limited. Although the manuscript alludes to estrogen receptor signaling and carcinogenesis, it fails to elaborate on molecular pathways such as estrogen metabolism (e.g., CYP450 enzymes), receptor subtypes (ERα vs ERβ), downstream gene targets, or cross-talk with growth factor signaling. Incorporating recent evidence from molecular oncology would strengthen the review.
  4. The discussion on hormone replacement therapy and oral contraceptives is overly generalized. Differentiating between estrogen-only vs combined regimens, dosage, duration, and patient stratification (e.g., age, menopausal status, BRCA mutation) is essential to present an accurate risk profile.
  5. The section on environmental estrogens (e.g., xenoestrogens, EDCs) should be more nuanced. Not all exogenous estrogens have the same potency, bioavailability, or risk profiles. Some compounds (e.g., phytoestrogens) may even exert protective or biphasic effects depending on context, which is not discussed here.
  6. There is a tendency toward uncritical summarization. The manuscript often cites studies without evaluating their methodological quality, consistency, or translational value. A table summarizing key studies, their design, main findings, and limitations would be beneficial for readers.
  7. The authors should adopt a more balanced tone. Several statements are overly speculative or alarmist (e.g., “estrogens in food are ignored threats”), which detracts from the scientific rigor of the review. A more measured, evidence-based language would improve credibility.
  8. Recent publications (within the last 5 years) are underrepresented. A review article must demonstrate up-to-date scholarship. Integrating current systematic reviews, molecular oncology findings, or EDC toxicology reports would help anchor the review in contemporary science.
  9. Figures or visual summaries are lacking. A schematic diagram showing the sources of exogenous estrogens, their molecular targets, and the downstream oncogenic effects would enhance clarity and impact.
  10. The conclusion needs to be more actionable. Rather than simply restating concerns, the authors should outline specific knowledge gaps, future research directions (e.g., exposure assessment, epigenetic effects), or clinical/public health implications (e.g., screening, regulation, education).

Author Response

REVIEWER 5: COMMENTS AND SUGGESTIONS FOR AUTHORS

GENERAL COMMENTS OF REVIEWER 5

This manuscript provides a narrative review of the current understanding of how exogenous estrogens, derived from pharmaceuticals, environmental contaminants, dietary sources, and other external exposures, may contribute to breast cancer risk. The topic is timely and of clinical and public health relevance. However, the review suffers from a lack of depth, limited critical synthesis of the literature, and insufficient mechanistic or epidemiological detail. The manuscript currently reads as a general overview rather than a high-impact, analytical review expected by Cancers

OUR RESPONSE

We have tried our best to improve the manuscript and represented that environmental estrogens pose a serious threat to the development of breast and other cancers. The manuscript is already large in length, since we have chosen all forms of exogenous estrogens, including pharmaceutical estrogens, xenoestrogens, and phytoestrogens, in a single manuscript. Therefore, there is no further scope for the in-depth analysis of any single topic.

REVIEWER 5, COMMENT 1: The manuscript lacks a clear organizing structure or conceptual framework. A more rigorous breakdown, such as by type of exposure (e.g., pharmaceutical, environmental, dietary), mechanism of action (e.g., ER-dependent vs independent), or level of evidence (e.g., in vitro, in vivo, clinical, epidemiological), would greatly improve readability and coherence.

OUR RESPONSE

Again, this field is very large. It needs several manuscripts to cover all the areas. We have already published a few manuscripts depicting the role of ERs in pro-oxidative and anti-oxidative actions of estrogens and on other related topics on estrogen. Since the manuscript is already very large, there is no scope to further enlarge it.

REVIEWER 5 COMMENT 2: Key landmark studies and meta-analyses are either missing or superficially mentioned. For instance, large-scale epidemiological studies from WHI, EPIC, or Nurses’ Health Study, which significantly shaped current understanding of estrogen-related breast cancer risk, are not adequately reviewed or contextualized.

OUR RESPONSE

Now we have added both the Nurses’ Health Study (NHS) and Women’s Health Initiative (WHI) study.

The following modified paragraph is added in section 3.1 of the manuscript,

Additionally, low-dose tropical HRT can be administered as vaginal creams, tablets, and rings without increasing the BC chances [24]. The Nurses' Health Study (NHS) and its 10-year follow-up after adjusting for age and other risk factors claimed that the overall relative risk of major coronary disease in women currently taking estrogen was 0.56 (95 percent confidence interval, 0.40 to 0.80). The results indicate that the risk of coronary diseases was significantly reduced among women with either natural or surgical menopause (Stampfer MJ, 1991). Several studies have claimed that pharmaceutical estrogens used in hormone replacement therapy (HRT) and oral contraceptive pills increase breast cancer risk. Conversely, other studies argue that progesterone, not estrogen, in contraceptive pills is what elevates this risk. An eminent Women’s Health Initiative (WHI) effort claimed that systematic estrogen monotherapy does not increase BC risk, and perhaps, in certain conditions, it may even lower the BC risk. However, systemic combination therapy increases the BC risk, even after ten years of HRT use termination in females. Additionally, higher doses of combined HRT (estrogen and progestins) confer a greater BC risk via relapse in advanced BC patients [25, 26]. In a WHI randomized trial, estrogen + progesterone therapy and family history were screened as independent (non-interacting) factors towards invasive BC risk amongst the sufferers [27].

New references added

M.J. Stampfer, GA Colditz, W.C. Willett, JE Manson, B. Rosner, F.E. Speizer, C.H. Hennekens. Postmenopausal estrogen therapy and cardiovascular disease. Ten-year follow-up from the nurses' health study. N Engl J Med. 1991 Sep 12;325(11):756-62.

REVIEWER 5, COMMENT 3: The mechanistic insights are limited. Although the manuscript alludes to estrogen receptor signaling and carcinogenesis, it fails to elaborate on molecular pathways such as estrogen metabolism (e.g., CYP450 enzymes), receptor subtypes (ERα vs ERβ), downstream gene targets, or cross-talk with growth factor signaling. Incorporating recent evidence from molecular oncology would strengthen the review.

OUR RESPONSE

We have added the role of ERα vs ERβ throughout wherever it is possible in the manuscript. For example,

Based on the other reviewer's response, we have added a paragraph (paragraph 4.6) representing the crosstalk with other signalling molecules, such as growth factors, etc.

REVIEWER 5, COMMENT 4: The discussion on hormone replacement therapy and oral contraceptives is overly generalized. Differentiating between estrogen-only vs combined regimens, dosage, duration, and patient stratification (e.g., age, menopausal status, BRCA mutation) is essential to present an accurate risk profile.

OUR RESPONSE

Based on the reviewer’s comments, we have revisited the sections (3.1) and added the word dose response wherever it is mentioned in the related publication.

REVIEWER 5, COMMENT 5: The section on environmental estrogens (e.g., xenoestrogens, EDCs) should be more nuanced. Not all exogenous estrogens have the same potency, bioavailability, or risk profiles. Some compounds (e.g., phytoestrogens) may even exert protective or biphasic effects depending on context, which is not discussed here.

OUR RESPONSE

In the manuscript it is never mentioned that all the exogenous estrogens have the same potency etc.

The concentration dependent response of the phytoesteogensare mentioned in the following subsection,

5.3. Concentration-dependent response of phytoestrogens on breast cancer complications

REVIEWER 5, COMMENT 6: There is a tendency toward uncritical summarization. The manuscript often cites studies without evaluating their methodological quality, consistency, or translational value. A table summarizing key studies, their design, main findings, and limitations would be beneficial for readers.

OUR RESPONSE

In the revised manuscript, we have tried to include and discuss the studies in an improvised manner. Attempts have been made to complement the observations with basic information, such asteh inclusion of Table 1 and improving Table 3. Besides, the information about various sources of estrogens and the molecular targets of estrogens for physiological access have been depicted through two new figures, responded in your Comment Number 9 separately.

REVIEWER 5, COMMENT 7: The authors should adopt a more balanced tone. Several statements are overly speculative or alarmist (e.g., “estrogens in food are ignored threats”), which detracts from the scientific rigor of the review. A more measured, evidence-based language would improve credibility.

OUR RESPONSE

We have tried to correct the sentences wherever we thought loose, replacing the speculative words throughout.

REVIEWER 5, COMMENT 8: Recent publications (within the last 5 years) are underrepresented. A review article must demonstrate up-to-date scholarship. Integrating current systematic reviews, molecular oncology findings, or EDC toxicology reports would help anchor the review in contemporary science.

OUR RESPONSE

There are 26 studies discussed in the manuscript, which were published within the last five years. In the revised version, we have added 16 new studies which have been published within the last five years. So, a total of 42 references are from the last five years.

REVIEWER 5, COMMENT 9: Figures or visual summaries are lacking. A schematic diagram showing the sources of exogenous estrogens, their molecular targets, and the downstream oncogenic effects would enhance clarity and impact.

OUR RESPONSE

As mentioned by the reviewer, two new figures have been added to the manuscript. These are on the sources of exogenous estrogens and the interacting targets for physiological entry of exogenous estrogens. Following are these figures, added as Fig.7 and Fig.9.

REVIEWER 5, COMMENT 10: The conclusion needs to be more actionable. Rather than simply restating concerns, the authors should outline specific knowledge gaps, future research directions (e.g., exposure assessment, epigenetic effects), or clinical/public health implications (e.g., screening, regulation, education).

OUR RESPONSE

We really feel worthy for this suggestion and as also suggested by Reviewer 1, a completely new conclusion (as under) is now included in the manuscript:

  1. Conclusions and future perspectives

  BC is the most common cancer worldwide and the second leading cause of cancer-related deaths. In premenopausal women, although the ovaries are the major sources of estrogens, adipose tissue and adrenal glands contribute small amounts of estrogens. Menopause significantly decreases estrogen production from the ovaries. However, adipose tissue, adrenal glands, and tissues such as the brain and lungs continue to produce estrogens due to the expression of the aromatase enzyme at the local tissue level. Besides natural estrogens, the human body, including both pre- and postmenopausal women, regardless of age and menstrual status, is exposed to various pharmaceutical estrogens, environmental xenoestrogens, and food-derived phytoestrogens. Many of these compounds, whether pharmaceutical estrogens, xenoestrogens, or phytoestrogens, bind to ERs with different affinities and cause diverse actions. At the physiological level, the interaction between estrogen and ERs regulates reproductive functions as well as other non-reproductive processes. Under certain conditions, natural, synthetic, or environmental estrogens can trigger neoplastic transformations in human epithelial cells, leading to cancers of the breast, ovaries, uterine endometrium, and lungs.

  Therefore, when using pharmaceutical estrogens for therapeutic purposes in premenopausal women, caution is necessary, and consulting a physician is strongly recommended, especially regarding dose, duration, and type of estrogen administered. Physicians' recommendations and advice are highly essential regarding the dose of hormonal therapy (low dose versus high dose) and whether monotherapy (estrogen-only therapy) or combination therapy (estrogen + progesterone) may be used. Premenopausal women are particularly vulnerable to the effects of pharmaceutical estrogens due to the high physiological levels of estrogen produced by their ovaries. In postmenopausal women, endogenous estrogen synthesis sharply decreases, leading to symptoms such as osteoporosis, vaginal dryness, and increased risks of CVDs and other conditions. Several studies, including the Nurses' Health Study (NHS), Women’s Health Initiative (WHI), and others, show the benefits of hormone replacement therapy (HRT) in alleviating postmenopausal symptoms. However, other research contradicts these benefits and warns against continued estrogen use because of potential cancer risks and adverse effects on the cardiovascular system. Besides natural and synthetic estrogens, the human body is exposed to environmental xenoestrogens from sources such as pesticides, plastic containers, and cosmetics, which can disrupt the physiology of endogenous estrogens and thus are regarded as endocrine-disrupting chemicals (EDCs). Precautions should be taken to minimize exposure to xenoestrogens due to their ability to cause genetic and epigenetic modifications (particularly DNA methylation and acetylation) and promote the development and progression of various cancers. Many studies suggest that pro-carcinogenic xenoestrogens tend to bind strongly to ERα, promoting cancer development.

  In contrast, phytoestrogens, a group of diverse compounds found in foods such as soy products, may offer health benefits or even protection against the development of cancers. Diverse mechanisms of protective action of phytoestrogens are observed, including antioxidative and anti-inflammatory actions. Additionally, a comparative study by Gustafson and his team (the discoverer of ERβ) showed that 17β-estradiol, the most abundant natural estrogen in women, has nearly equal affinity for ERα and ERβ. On the other hand, phytoestrogens like genistein, derived from soy, exhibit ~400 times greater affinity for ERβ, which contributes to their protective effects against cancers and CVDs. Other studies noted that the protective effects of phytoestrogens on human health, including their role in preventing BCs, depend on factors such as ethnicity (e.g., South-Southeast Asian people), although the majority of the study results claimed that it is the specific food habits, rather than genetics which takes a decisive role of the protective actions of phytoestorgens. Although several clinical studies have explored the potential benefits of phytoestrogens, conclusive evidence confirming their role in cancer prevention is still lacking. The patient-derived xenograft models and nanoparticle-dependent drug delivery are important areas of present-day research in breast cancer therapy (Li J, 2025; Ma P, 2024; Huang M, 2023). More research is needed on the impact of environmental estrogens/phytoestrogens, including the effects of combination exposure, keeping in mind the sex, age, and other critical factors, which would further enhance our knowledge in this area.

Reviewer 2: Recommended New References are added here

1] J. Li, A. Gu, N. Tang, G. Zengin, M.-Y. Li, Y. Liu, Patient-derived xenograft models in pan-cancer: From bench to clinic. Interdiscip. Med. 2025, 3, e20250016. DOI: 10.1002/INMD.20250016

[2] Huang M, Gong G, Deng Y, Long X, Long W, Liu Q, et al. Crosstalk between cancer cells and the nervous system. Med Adv. 2023; 1(3): 173-189. https://doi.org/10.1002/med4.27

[3] P. Ma, G. Wang, K. Men, et al. Advances in clinical application of nanoparticle-based therapy for cancer treatment: A systematic review, Nano TransMed. 3 (2024) 100036. https://doi.org/10.1016/j.ntm.2024.100036

Round 2

Reviewer 1 Report

Comments and Suggestions for Authors

thank you for the answers

Author Response

Dear Sir

Many thanks for your valuable assessment and providing knowledgeable inputs for enhancing the scientific rigor of our manuscript. Your constructive feedback and evaluation kept us focused and encouraging throughout.

Sincerely  

Tapan Kumar Mukherjee

Professor in Biotechnology

Amity University, Kolkata, West Bengal, India

Reviewer 3 Report

Comments and Suggestions for Authors

The authors addressed some reviewer comments; they need to respond to the remaining comments.

While the current review primarily focuses on the chemical properties of the compounds under investigation, it falls short in offering a thorough explanation of how each class of compounds interacts with biological systems. It is imperative that the authors include a comprehensive overview of the mechanisms of action for these exogenous estrogens. This should encompass relevant mechanistic pathways, ideally supplemented with an illustrative figure to visually represent these processes.

Moreover, the authors need to address the specific dosage levels at which these exogenous estrogens may pose a cancer risk. It is crucial to clearly distinguish between the therapeutic and clinical uses of these compounds, specifying the associated doses that may increase the risk of breast cancer. This distinction will provide clarity on the implications of their findings and guide future research and clinical practices.

Author Response

REVIEWER NO 3 COMMENT NUMBER 1:

While the current review primarily focuses on the chemical properties of the compounds under investigation, it falls short in offering a thorough explanation of how each class of compounds interacts with biological systems. It is imperative that the authors include a comprehensive overview of the mechanisms of action for these exogenous estrogens. This should encompass relevant mechanistic pathways, ideally supplemented with an illustrative figure to visually represent these processes.

OUR RESPONSE:

We have added the following new paragraph with a new figure:

 4.6. Mechanism of action of xenoestrogens in the development of cancers

 Exogenous estrogens, especially xenoestrogens, have enduring pathophysiological effects by either directly affecting genetic and epigenetic changes or by competing with endogenous estrogens for binding to estrogen receptors (ERs), thereby disrupting the natural hormonal signaling mediated by the estrogen-ER complex. Multiple lines of evidence indicate that both the normal physiological and pathological roles of endogenous estrogens are influenced by exogenous estrogens. This imbalance in estrogen levels caused by external xenoestrogen compounds impacts several pathological processes, including infertility, immune disorders, various cancers, and other health problems. A detailed examination of cellular and molecular events related to the development and progression of different cancers reveals that interaction between exogenous estrogens and ERs can initiate and promote multiple cancers, including breast cancer (BC), through four main signaling pathways. 1. In the classical genomic pathway, xenoestrogens diffuse into the cytoplasm and bind to dimeric ERs, then translocate to the nucleus. Here, the xenoestrogen-ER complex binds to estrogen response elements (ERE) in gene promoters controlled by estrogen, regulating transcription. However, not all estrogen-responsive gene promoters contain EREs; some have regions where specific transcription factors bind. 2. In the non-classical genomic pathway, when EREs are absent, the cytoplasmic xenoestrogen-ER complex activates various transcription factors (e.g., SP1, AP1, GATA, STAT) which then move to the nucleus, bind to specific regions in gene promoters, and regulate transcription. 3. The third pathway involves membrane-bound ERs (mERs) and G-protein coupled estrogen receptors (GPERs), responsible for rapid, non-genomic actions, such as activating ion channels on the cell membrane. These receptors activate transcription factors that eventually bind to gene promoters of estrogen-activating genes to trigger transcription. 4. The fourth pathway describes how estrogen response elements (ERE) in estrogen-responsive genes can be activated without estrogens, by the binding of specific growth factors (GFs) to membrane growth factor receptors (GFRs). The GF-GFR complex activates kinases that lead to phosphorylation of ERs, thus activating them and allowing their binding to EREs. Several other regulators and co-activators of ERE have been identified, including aryl hydrocarbon receptors (AhRs), peroxisome proliferator-activated receptors (PPARs), and reactive oxygen species (ROS). Additionally, when activated, estrogen-related receptors (ERRs), a group of orphan nuclear receptors, bind to ERREs of specific genes and regulate their transcription. Xenoestrogen signaling activates several proto-oncogenes, turning them into oncogenes, and inactivates tumor suppressor genes along with genes involved in inflammation and oxidative stress. Key cell signaling molecules and pathways affected by xenoestrogens include those involved in cell proliferation (e.g., cyclin D), survival (e.g., AKT), apoptosis (e.g., Bcl-2), angiogenesis (e.g., VEGF/VEGFR), and invasion and metastasis (e.g., MMPs) in cancer cells. In a comprehensive review, Wang X et al. described the impact of xenoestrogens and phytoestrogens on various cell signaling molecules and cellular events related to normal and abnormal functions [111 Wang X, 2021]. A schematic diagram depicts the various mechanisms through which xenoestrogens mediate pro-carcinogenic action (Figure 12)

New references added

Wang, X.; Ha, D.; Yoshitake,R.; Chan, Y.S.; Sadava, D.; Chen, S. Exploring the Biological Activity and Mechanism of Xenoestrogens and Phytoestrogens in Cancers: Emerging Methods and Concepts. Int. J. Mol. Sci. 2021, 22, 8798. https://doi.org/10.3390/ijms22168798.

Figure 12 is newly added as per your worthy suggestions, depicting the various mechanisms by which exogenous estrogens could aggravate the BC pathogenesis.

REVIEWER NO 3 COMMENT NUMBER 3

Moreover, the authors need to address the specific dosage levels at which these exogenous estrogens may pose a cancer risk. It is crucial to clearly distinguish between the therapeutic and clinical uses of these compounds, specifying the associated doses that may increase the risk of breast cancer. This distinction will provide clarity on the implications of their findings and guide future research and clinical practices.

 OUR RESPONSE:

We have already discussed in detail the effects of low-dose and high-dose of pharmaceutical estrogens and phytoestrogens and dedicated specific paragraphs. We have also discussed the Nurses’ Health Study, WHI, etc to show the effects of low dose versus high dose estrogen as a monotherapy and as a combination therapy with progesterone. For xenoestrogens, we are adding the following paragraph at the end of page 12.:

Traditionally, it is believed that the adverse effects of xenoestrogens are proportional to their doses, and no adverse effects are seen at exposures below the LOAEL (lowest observed adverse effect level) (Waddell, 2010). However, different hazard assessment endpoints are studied in low-dose versus high-dose research, and a non-monotonic dose-response (NMDR) curve is created (Beausoleil et al., 2013; Viñas and Watson, 2013). These findings strongly suggest that not only a single, but also a mixture of XEs at low doses can significantly affect cell proliferation, apoptosis, and tissue development even at extremely low concentrations (e.g., picomolar) (Xu Z, 2017).

New references added

  1. Waddell, W.J., 2010. History of dose response. J. Toxicol. Sci. 35, 1–8.
  2. Beausoleil, C., et al., 2013. Low dose effects and non-monotonic dose responses for endocrine active chemicals: science to practice workshop: workshop summary. Chemosphere 93, 847–856.
  3. Viñas, R., Watson, C.S., 2013. Mixtures of xenoestrogens disrupt estradiol-induced nongenomic signaling and downstream functions in pituitary cells. Environ. Health-Glob. 12, 26.
  4. Xu Z, Liu J, Wu X, Huang B, Pan X. Nonmonotonic responses to low doses of xenoestrogens: A review. Environ Res. 2017; 155:199-207

Reviewer 4 Report

Comments and Suggestions for Authors

The authors have made significant revisions to the manuscript and hence can be accepted.

Author Response

Honorable Sir

Sincere thanks for your worthy evaluation and constructive feedback on our manuscript.

Tapan Kumar Mukherjee

Professor in Biotechnology

Amity University, Kolkata, West Bengal, India

Reviewer 5 Report

Comments and Suggestions for Authors

The authors have made commendable efforts to improve the manuscript. The inclusion of WHI/NHS data, recent references, revised conclusion, and new figures are appreciated. However, a few key issues remain:

  1. The manuscript would benefit from a clearer organizational framework (e.g., by exposure type or mechanism) to improve coherence.
  2. Please expand briefly on estrogen metabolism (e.g., CYP450), ER subtypes, and downstream targets for a more robust mechanistic overview.

Author Response

COMMENTS AND SUGGESTIONS OF REVIEWER NO. 5 AND OUR RESPONSE

The authors have made commendable efforts to improve the manuscript. The inclusion of WHI/NHS data, recent references, revised conclusion, and new figures is appreciated. However, a few key issues remain:

 REVIEWER NO 5 COMMENT NUMBER 1:

The manuscript would benefit from a clearer organizational framework (e.g., by exposure type or mechanism) to improve coherence.

OUR RESPONSE

We have worked very hard and dedicated an extensive amount of time to write and arrange this manuscript. It is not possible for us to rearrange the whole manuscript at this time.

 REVIEWER NO 5 COMMENT NUMBER 2:

Please expand briefly on estrogen metabolism (e.g., CYP450), ER subtypes, and downstream targets for a more robust mechanistic overview.

 OUR RESPONSE

Very briefly, we have added the role of CYP enzymes in estrogen metabolism at the end of page 4 and the beginning of page 5.

Estrogens undergo extensive modifications, primarily in the liver, through hydroxylation by several isoforms of cytochrome P450 (CYP) enzymes, namely CYP1A1, CYP1A2, CYP1B2, and CYP3A4.  These enzymes catalyse the estradiolin 2-, 4-, and 16α-positions, generating 2-hydroxyestradiol (2-OH-E2), 4-hydroxyestradiol (4-OH-E2), and 16α-hydroxyestrone (16α-OH-E1). The catechol estrogens are further oxidized to form DNA-damaging semiquinones and quinones. Additionally, following CYP-mediated hydroxylation, estrogens can undergo further metabolism through glucuronidation, sulfation, and O-methylation(Tsuchiya Y, 2005).

New references added

 Tsuchiya Y, Nakajima M, Yokoi T. Cytochrome P450-mediated metabolism of estrogens and its regulation in human. Cancer Lett. 2005 Sep 28;227(2):115-24.

 For ER subtypes, and downstream targets for a more robust mechanistic overview comments.

Based on the reviewer 3 comments we have added details mechanisms of action of ERs and estrogen in the section 4.6 of the manuscript with a new figure. Please check the response and the new figure we have added in response to reviewer 3.

Round 3

Reviewer 3 Report

Comments and Suggestions for Authors

The authors made significant revisions based on the reviewers' comments, and the manuscript can be acceptable for publication.